# Analysis of cardiac magnetic resonance imaging in 36,000 individuals yields genetic insights into dilated cardiomyopathy

James P. Pirruccello [1,2,3], Alexander Bick[2,3,4], Minxian Wang [3], Mark Chaffin [3], Samuel Friedman[5], Jie Yao[6], Xiuqing Guo[6], Bharath Ambale Venkatesh[7], Kent D. Taylor[6], Wendy S. Post[8,9], Stephen Rich [10], Joao A. C. Lima [11,7], Jerome I. Rotter[6], Anthony Philippakis[5,12], Steven A. Lubitz[1,2,3,13], Patrick T. Ellinor[1,2,3,13], Amit V. Khera[1,2,3,13], Sekar Kathiresan [1,2,3,13,14] & Krishna G. Aragam[1,2,3,13 ✉]

Dilated cardiomyopathy (DCM) is an important cause of heart failure and the leading indication for heart transplantation. Many rare genetic variants have been associated with DCM, but common variant studies of the disease have yielded few associated loci. As structural changes in the heart are a defining feature of DCM, we report a genome-wide association study of cardiac magnetic resonance imaging (MRI)-derived left ventricular measurements in 36,041 UK Biobank participants, with replication in 2184 participants from the Multi-Ethnic Study of Atherosclerosis. We identify 45 previously unreported loci associated with cardiac structure and function, many near well-established genes for Mendelian cardiomyopathies. A polygenic score of MRI-derived left ventricular end systolic volume strongly associates with incident DCM in the general population. Even among carriers of *TTN* truncating mutations, this polygenic score influences the size and function of the human heart. These results further implicate common genetic polymorphisms in the pathogenesis of DCM.

[1] Division of Cardiology, Massachusetts General Hospital, Boston, MA, USA. [2] Center for Genomic Medicine, Massachusetts General Hospital, Boston, MA, USA. [3] Program in Medical and Population Genetics, Broad Institute, Cambridge, MA, USA. [4] Department of Medicine, Massachusetts General Hospital, Boston, MA, USA. [5] Data Sciences Platform, Broad Institute, Cambridge, MA, USA. [6] The Institute for Translational Genomics and Population Sciences, Department of Pediatrics, Los Angeles Biomedical Research Institute at Harbor-UCLA Medical Center, Torrance, CA, USA. [7] Department of Radiology, Johns Hopkins University, Baltimore, MA, USA. [8] Department of Epidemiology, Johns Hopkins University Bloomberg School of Public Health, Baltimore, MA, USA. [9] Department of Medicine, Division of Cardiology, Johns Hopkins University School of Medicine, Baltimore, MA, USA. [10] Center for Public Health Genomics, University of Virginia, Charlottesville, VA, USA. [11] Division of Cardiology, Johns Hopkins University School of Medicine, Baltimore, MA, USA. [12] GV, Mountain View, CA, USA. [13] Harvard Medical School, Boston, MA, USA. [14] Verve Therapeutics, Cambridge, MA, USA. ✉email: karagam@mgh.harvard.edu

Affecting one in every 250 people, dilated cardiomyopathy (DCM) is a disease of cardiac muscle that leads to heart failure, and is the most common indication for cardiac transplantation[1]. Rare variants in dozens of genes have been associated with DCM[2]. The most commonly identified mutations are truncating variants in *TTN* (TTNtv), which are found in 15–20% of DCM cases. However, rare variants in cardiomyopathy-related genes yield a genetic diagnosis for DCM in approximately 40% of cases[2]. Furthermore, sequencing efforts in large populations are now identifying an increasing number of individuals who harbor rare DCM-associated variants, but do not manifest clinical disease[3–6].

A putative explanation for the limited diagnostic yield and incomplete penetrance of rare DCM-associated variants is that common genetic variation plays a role in the genetic architecture of DCM. To gain insight into the relationship between common genetic variants and DCM, one exome-wide association study[7] and two genome-wide association studies (GWAS)[8,9] have been conducted. These case–control studies identified nine loci significantly associated with DCM, five of which contain genes that also harbor rare DCM-causing mutations (*TTN*, *ALPK3*, *BAG3*, *FLNC*, and *PLEKHM2*). These studies have successfully linked common variants to DCM, though their yield has been limited by modest sample sizes, as each study had fewer than 5,000 cases.

As structural changes to the left ventricle are a defining—and frequently incipient—feature of DCM, genetic analyses of cardiac imaging traits present another plausible method for genetic discovery[10,11]. Several GWAS of cardiac imaging phenotypes have probed the link between common genetic variants and changes in cardiac structure and function. For example, a large-scale genetic study of over 46,000 participants with transthoracic echocardiography (TTE)-derived phenotypes identified five genome-wide significant loci. Notably, one locus, near *PLN*, is associated with Mendelian cardiomyopathies[12,13]. A subsequent study using TTE in 19,000 participants from BioBank Japan yielded five additional, previously unreported loci, including a locus near *VCL* also associated with cardiomyopathies[14–16]. Finally, a study of 6765 African-American participants with cardiac imaging from the Candidate Gene Association Resource (CARe) Study, which included 1210 individuals with cardiac magnetic resonance imaging (MRI) from the Multi-Ethnic Study of Atherosclerosis (MESA), yielded four suggestive loci that have not been subsequently confirmed[17]. Together, these studies highlight the potential value of population-based cardiac imaging to gain clinical and biological insights into myocardial diseases. Whether common genetic variants related to the heart's structure and function can identify as-yet unaffected individuals at risk for DCM remains uncertain.

In this study, we investigate common variant associations with cardiac structure and function in the UK Biobank, a population-based cohort of over 500,000 participants[18,19], including a large imaging substudy with plans to perform cardiac MRI in 100,000 participants[20,21]. The phenotypic characteristics of the first 5000 participants have been described in detail[22,23]. Here, we analyze automated measurements from 36,041 participants in order to study the relationship between common genetic variants, cardiac imaging phenotypes, and risk for the development of DCM. We identify 45 previously unreported loci associated with left ventricular structure and function, demonstrate substantial overlap between the uncovered genetic loci and known Mendelian cardiomyopathy genes, and develop a robust polygenic predictor of incident DCM.

## Results

**Phenotype refinement and cardiac MRI results**. We identified 36,041 UK Biobank participants with cardiac MRI readings provided by the UK Biobank who did not have a diagnosis of congestive heart failure (CHF), coronary artery disease (CAD), or DCM at the time of enrollment (Supplementary Data File 1 and Supplementary Fig. 1). For these individuals, seven cardiac MRI-derived phenotypes were available: left ventricular end-diastolic volume (LVEDV), left ventricular end-systolic volume (LVESV), stroke volume (SV), the body-surface-area (BSA) indexed versions of each of these traits (LVEDVi, LVESVi, and SVi), and left ventricular ejection fraction (LVEF). At the time of MRI, the individuals had a mean age of 64 years and the majority were female (52.9%). For women, the mean LVEDV, LVESV, and SV were 120, 41, and 81 mL, respectively; for men, they were 150, 58, and 95 mL (Supplementary Data File 2). The LVEF averaged 67% for women and 63% for men (Supplementary Fig. 2).

**Cardiac structure and function are heritable**. We asked whether the cardiac MRI phenotypes were influenced by participants' genetic backgrounds. SNP-based heritability—the proportion of variance explained by all SNPs on a genotyping array—was 43% for LVEDV, 40% for LVESV, 31% for LVEF, and 34% for SV. Genetic correlations between these cardiac MRI phenotypes were similar in magnitude to their observational correlations, except for the relationship between SV and LVEF, which had a near-zero observational correlation but a weakly positive genetic correlation (Supplementary Figs. 3 and 4).

**Fifty-seven genetic loci associated with cardiac size and function**. Having established a genetic basis for variability in cardiac structure and function, we then performed a series of GWAS to identify common genetic variants associated with the seven cardiac MRI phenotypes. Fifty-seven distinct loci in the human genome were associated with at least one cardiac MRI phenotype at a widely used genome-wide significance threshold (BOLT-LMM $P < 5 \times 10^{-8}$; Supplementary Data File 3). If a more stringent threshold of $P < 5 \times 10^{-9}$ had been used, we would have identified 36 distinct loci (Supplementary Table 1). Linkage disequilibrium (LD) score regression revealed minimal test statistic inflation, consistent with polygenicity rather than population stratification (Supplementary Table 2). No lead SNP deviated from Hardy–Weinberg equilibrium (HWE) beyond the threshold of HWE $P = 1 \times 10^{-6}$.

Of the 57 genome-wide significant loci, 45 were previously unreported and had not been described in prior common variant analyses of DCM or cardiac imaging phenotypes. In total, 22 loci associated with LVEDV, 14 with LVEDVi, 32 with LVESV, 28 with LVESVi, 22 with LVEF, 12 with SV, and eight with SVi (Fig. 1).

Five of the ten loci previously discovered through genetic analyses of echocardiographic measurements were among our genome-wide significant loci, including SNPs in loci near *PLN*, *SH2B3/ATXN2*, *MTSS1*, *SMARCB1/DERL3*, and *CDKN1A*[13,14]. The comparison to the echocardiographic studies was performed on a per-trait basis, considering fractional shortening on TTE to be analogous to LVEF on MRI, left ventricular internal diameter at end diastole to be analogous to LVEDV, and left ventricular internal diameter at end systole to be analogous to LVESV. Of the four loci previously identified in a GWAS of cardiac traits measured with TTE in participants of European ancestry[13], all four were also significantly associated with an analogous trait in our study (Supplementary Table 3). In comparison, two of six loci identified in a GWAS of TTE-derived cardiac traits in participants of Japanese ancestry were significantly associated in our study[14], a lower proportion that may reflect ancestry-specific patterns of linkage disequilibrium.

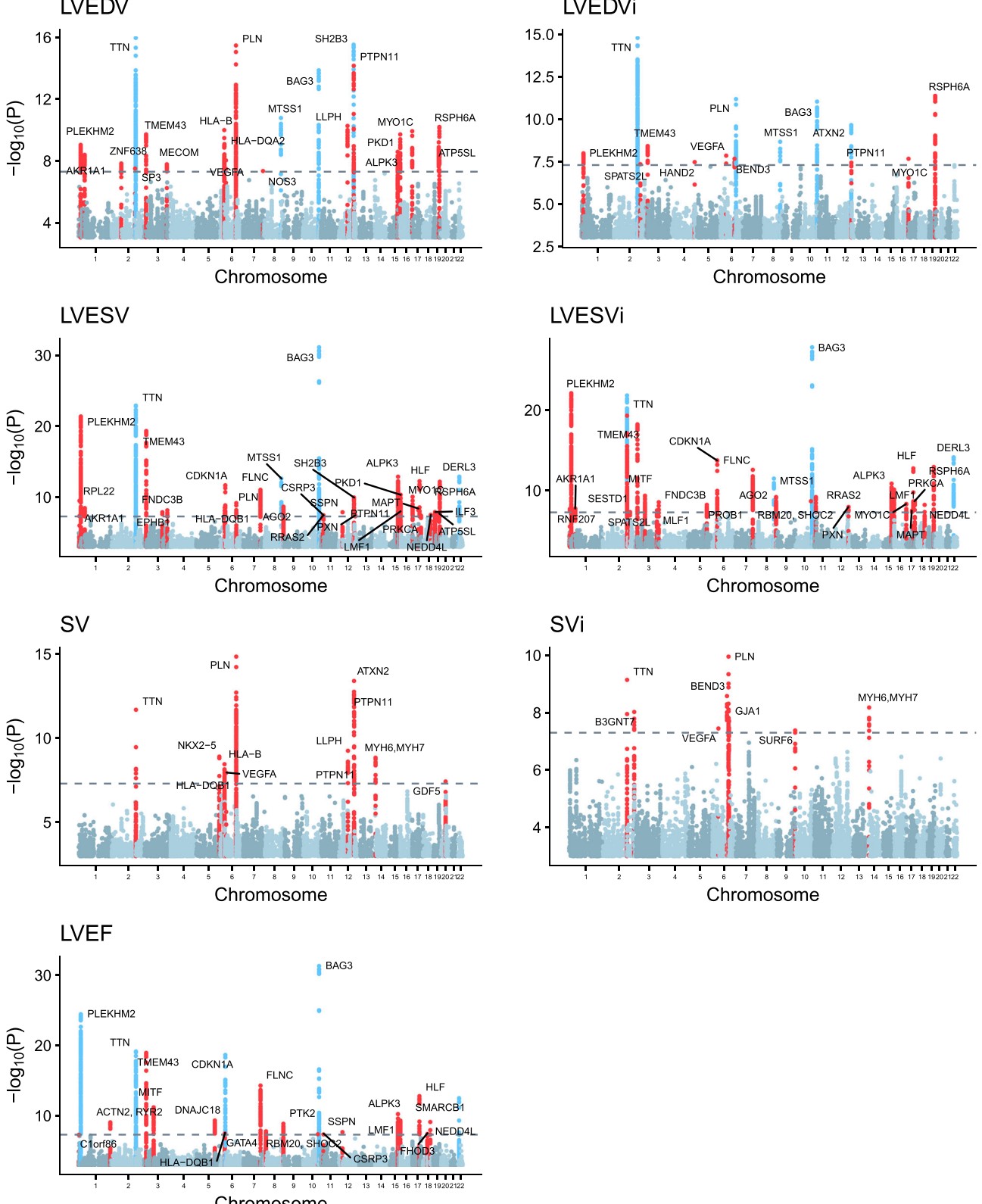

**Fig. 1 Manhattan plots of genome-wide association discovery analyses of cardiac MRI phenotypes.** For each cardiac MRI phenotype, the −log10(P value) is graphed on the y axis at each chromosomal position on the x-axis. P is the BOLT-LMM P value. The nearest gene to each genome-wide significant lead SNP is labeled at each locus, except when a cardiomyopathy-related gene is present within 500 kb of the lead SNP. SNPs are colored blue near loci that have previously been observed in common genetic analyses using cardiac traits (TTE or cardiac MRI). SNPs are colored red near loci that were previously unreported.

Six of the nine SNPs previously identified at exome- or genome-wide significance in common variant studies of cardiomyopathy were also associated at genome-wide significance with at least one cardiac MRI phenotype in our study, including SNPs near ALPK3, BAG3, CLCNKA/HSPB7, FHOD3, FLNC, and TTN[7–9]. Our data provide nominal supporting evidence (BOLT-LMM $P < 1 \times 10^{-3}$) for SNPs at two of the remaining three loci. The lead SNPs from these prior studies, as well as the association $P$ value from the most strongly associated cardiac MRI phenotype from our study, are available in Supplementary Table 3.

**External validation**. We pursued replication of our genome-wide significant loci in two independent cohorts (MESA and BioBank Japan) using two different cardiac imaging modalities (cardiac MRI and TTE).

In MESA, we studied 2184 participants with whole-genome sequencing and cardiac MRI data available, and who did not have cardiovascular disease or late gadolinium enhancement consistent with myocardial scar (Supplementary Table 4)[24–26]. Of the 138 genome-wide significant SNP-trait associations in our discovery analysis (accounting for SNPs significantly associated with multiple traits), 113 were available for comparison in the MESA dataset. In total, 99 of 113 SNP-trait associations had effects in the same direction for UK Biobank and MESA (87.6% concordance; binomial test with two-tailed $P = 6.1 \times 10^{-17}$, given a chance expectation of 50% at each of 113 sites). Of the 27 SNP-trait associations with a $P$ value < 0.05, 26 had an effect in the same direction as in the UK Biobank (binomial test with two-tailed $P = 5.4 \times 10^{-11}$, given a chance expectation of 5% at each of 113 sites). Validation results for each SNP are available in Supplementary Data File 4.

We then performed cross-modality validation using summary statistics from a genome-wide association study in BioBank Japan of 19,000 participants of Japanese ancestry with TTE data[14], including three traits analogous to LVEDV, LVESV, and LVEF. We were able to identify data from BioBank Japan for 46 SNP-trait associations (out of 76 in total across the three traits, accounting for sites associated with multiple traits). Of these 46 SNP-trait associations, 39 had an effect in the same direction as in the UK Biobank (84.8% agreement; binomial test with two-tailed $P = 1.8 \times 10^{-6}$; Supplementary Data File 4).

**Enrichment near genes expressed in cardiac and skeletal muscle**. In order to understand the tissue specificity of the discovered loci, we applied MAGMA, which revealed an enrichment of variants clustered near genes expressed in cardiac and skeletal muscle tissue types, with results for LVESV, LVEDVi, and LVESVi, achieving a Bonferroni-corrected significance threshold (Supplementary Fig. 5).

**GWAS loci enriched for Mendelian cardiomyopathy-linked genes**. We then asked whether the GWAS loci that we identified for cardiac structure and function were more enriched for known cardiomyopathy genes than expected by chance (Supplementary Table 5). We compared the likelihood of an overlap between known cardiomyopathy genes and our GWAS loci or matched control loci (see Methods for details). In 10,000 simulations, we found a significant enrichment in Mendelian cardiomyopathy-linked genes at the cardiac MRI GWAS loci (one-tailed permutation $P = 1 \times 10^{-4}$, Supplementary Fig. 6).

**Transcriptome-wide association study**. We performed a transcriptome-wide association study (TWAS) to prioritize genes within 500 kb of each locus based on expression in the left

ventricle and right atrial appendage[27,28]. TWAS-based prioritization is complementary to the approach of selecting the nearest gene at a locus[27,29]. The most strongly associated gene at each locus from the TWAS is annotated in Supplementary Data File 3. Seven of the cardiomyopathy-linked genes from Supplementary Table 6 (ACTN2, ALPK3, MYH6, NKX2–5, PLN, PTPN11, and SHOC2) were the genes most strongly prioritized by TWAS at their respective loci for at least one trait. Three cardiomyopathy-linked genes (BAG3, CSRP3, and TTN) were not candidates for inclusion in the TWAS, because they were below the inclusion threshold based on the genotype-tissue expression (GTEx) expression quantitative trait locus (eQTL) $P$ value in both the left ventricle and the right atrial appendage. Full TWAS results are displayed in Supplementary Data File 5.

**Polygenic score PheWAS yields a link to DCM**. We also produced polygenic scores for each trait, weighting the genetic dosage by the effect size of the lead SNPs from each GWAS in Supplementary Data File 3. We performed a cross-sectional phenome-wide association study (PheWAS) in 449,000 UK Biobank participants to assess the relationship between each of the seven polygenic scores and disease phenotypes. We first performed a PheWAS using a broad set of PheCodes[30], 1216 of which were present in 200 or more participants. As anticipated, this analysis showed an enrichment for cardiac diseases. We then performed a PheWAS for 96 diseases using curated definitions (as defined in Supplementary Data File 1). Among our curated disease traits, DCM emerged as the disease most strongly associated with the polygenic scores for LVEF, LVESV, and LVESVi. Among these, the LVESVi polygenic score had the single strongest relationship with DCM (OR 1.51 per standard deviation [SD] increase in LVESVi polygenic score; $P = 8.5 \times 10^{-34}$ by logistic regression, Fig. 2). The LVEDVi and SV polygenic scores had strong inverse relationships with hypothyroidism, an observation consistent with invasive studies that found reduced LVEDV and SV in hypothyroid patients, attributed to hemodynamic loading conditions[31]. The LVEDV and SV scores, both of which have contributions from the major histocompatibility complex (MHC), were most strongly inversely associated with psoriasis. Each of the seven polygenic scores had several significant associations with disease phenotypes, even after adjusting for multiple testing (Bonferroni-adjusted significance threshold $P = 5.9 \times 10^{-6}$ in the PheCode PheWAS, $P = 7.4 \times 10^{-5}$ in the manually curated disease phenotype PheWAS). The strongest relationships between each of the seven polygenic scores and the disease phenotypes are detailed in Supplementary Fig. 7. Manually curated disease phenotypes with Bonferroni-adjusted significant $P$ values are displayed in Supplementary Table 7; the full table of PheWAS results for PheCodes is available for download (Supplementary Data File 5).

**LVESVi polygenic score and incident DCM**. Having established a strong relationship between common variant-derived polygenic scores and DCM in the cross-sectional PheWAS, we then asked whether these polygenic scores also predicted incident disease. The association between polygenic scores and incident DCM (388 cases) was assessed in the remaining 358,556 individuals in the UK Biobank without cardiac MRI data, after excluding those with cardiac disease at baseline, and those with third-degree or closer relatedness to another participant (Supplementary Table 8). The polygenic scores for LVEDV, LVESV, LVEF, LVEDVi, and LVESVi were significantly associated with incident DCM after adjusting for age, sex, genotyping batch, and the first five principal components of ancestry in separate Cox proportional hazard models for each phenotype.

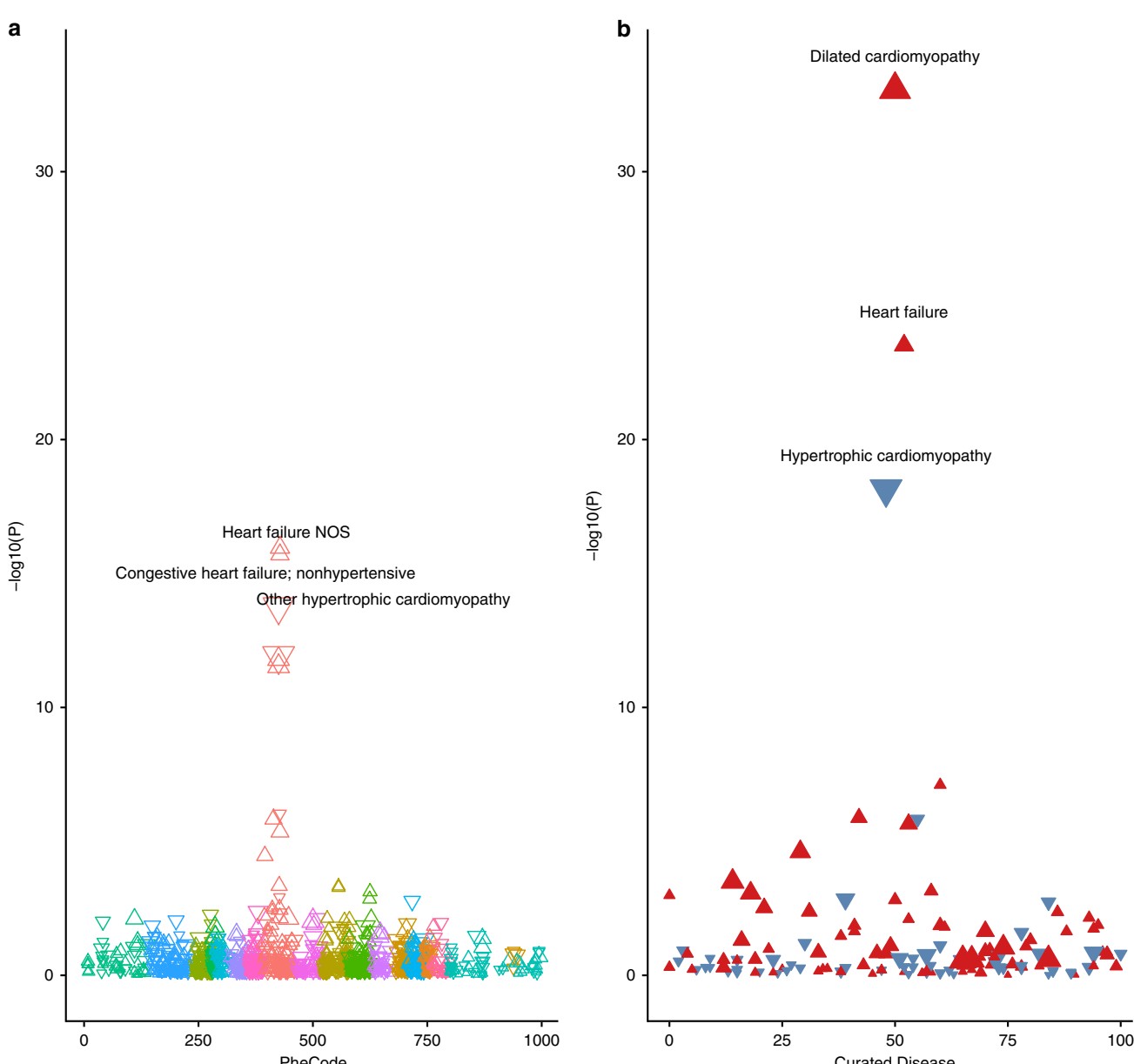

**Fig. 2 PheWAS highlights the connection between a polygenic score for LVESVi and dilated cardiomyopathy.** The polygenic score derived from LVESVi was applied to PheCodes (panel **a**) and curated disease phenotypes (panel **b**) in the UK Biobank. Each of the curated phenotypes is defined in Supplementary Table 1. For both panels (**a**) and (**b**), the x-axis represents the identifying code for the disease phenotype. The y-axis represents the −log10 of the P value of the association between the polygenic score and the phenotype in a logistic model adjusted for age at enrollment, the genotyping array, sex, and the first five principal components of ancestry. Triangles oriented upward represent betas that are concordant with the LVESVi PRS (e.g., a higher LVESVi PRS corresponds with a higher risk of DCM), and the reverse is true for downward-oriented triangles. The three most strongly associated phenotypes in each panel are labeled in the figure. The triangles in panel (a) are colored by three-digit PheCode. The triangles in panel (**b**) are colored red if positively correlated with the LVESVi polygenic score, and blue if negatively correlated. The PheWAS plots for all seven cardiac MRI phenotypes are available in Supplementary Fig. 7.

The 28-SNP LVESVi polygenic score had the strongest relationship with incident DCM (hazard ratio [HR] = 1.58 per SD increase in the score, $P = 6.4 \times 10^{-18}$ by Cox regression). The direction was consistent with clinical expectations: a greater genetically mediated LVESVi corresponded with a higher risk of DCM, while a lower polygenic risk corresponded to a lower risk of DCM. The cumulative incidence of DCM among those in the top 10%, bottom 10%, and middle 80% for the polygenic score is shown in Fig. 3. For each trait, the relationship between its

polygenic score and incident DCM is available in Supplementary Table 9.

**Influence of polygenic score in *TTN* truncation carriers.** Finally, to assess whether common genetic variants might contribute to the incomplete penetrance and variable expressivity of DCM-associated rare variants, we asked whether the 28-SNP LVESVi polygenic score affected the structure and function of the heart among carriers of truncating variants in *TTN* exons that are

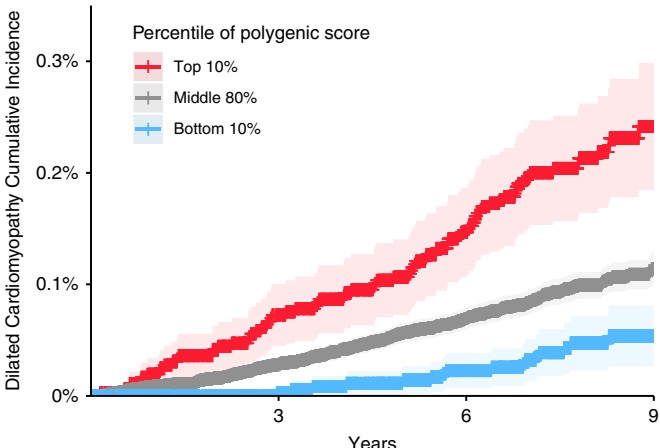

**Fig. 3 The LVESVi polygenic score influences the risk for incident dilated cardiomyopathy.** The cumulative DCM incidence (defined as 1 minus the Kaplan–Meier survival estimate) is plotted for individuals in the bottom tenth percentile (blue), middle 80% (gray), and top tenth percentile (red) for the LVESVi polygenic score. The 95% confidence intervals (derived from the standard error of the cumulative hazard) are represented with lighter colors. The x-axis represents the number of years since enrollment in the UK Biobank. The y-axis represents the cumulative incidence of DCM.

highly expressed in the heart (TTNtv). Rare TTNtv were chosen for analysis because of their established role in cardiomyopathy and preclinical cardiac dysfunction, as well as their relatively high population frequency (~0.5%). Among the 12,448 participants who had undergone both cardiac MRI and exome sequencing, we identified 59 carriers of TTNtv in exons previously shown to be spliced into over 90% of transcripts in the heart (Supplementary Table 10). Although we identified additional participants with imputation data that indicated that they were carriers of TTNtv, none of these were confirmed by sequencing (Supplementary Table 11), so we proceeded to analyze only the 59 participants with directly sequenced TTNtv. The polygenic background of TTNtv carriers influenced left ventricular volume and function: the LVESV of TTNtv carriers was influenced by the LVESVi polygenic score (7.2-mL increase in LVESV per SD, $P = 0.011$ by linear regression), and LVEF was also reduced by 2.6% per SD of the LVESVi score ($P = 0.006$ by linear regression). The LVEDV of TTNtv carriers was not significantly influenced by the LVESVi polygenic score after Bonferroni adjustment (7.9-mL increase in LVEDV per SD increase in the score, $P = 0.046$ by linear regression).

**Sensitivity analysis in European-ancestry participants.** To understand whether our results were confounded by residual population stratification, the primary genetic analyses (heritability assessment, GWAS, and polygenic score estimates) were repeated in a subset consisting of samples that were within a tight genetic inlier cluster among participants with self-reported European ancestry ($N = 32,755$). These sensitivity analyses yielded similar heritability estimates, fewer genome-wide significant loci (49 instead of 57), and similar polygenic score effect estimates when compared with the main analyses using all 36,041 samples (for details, see Supplementary Methods, Supplementary Notes 1, Supplementary Fig. 8, and Supplementary Tables 12–20).

## Discussion

In this study, we identified 45 previously unreported, common genetic loci associated with the structure and function of the heart in individuals without overt cardiovascular disease. We

then linked our identified genetic loci to DCM through multiple lines of evidence, established that a polygenic score derived from common variants predicts incident DCM, and demonstrated that a common genetic background affects cardiac traits even among carriers of cardiomyopathy-related rare variants.

These results permit several conclusions. First, common genetic variants that are associated with left ventricular structure and function contribute to DCM risk. In an incident disease analysis, our best-performing polygenic score—comprising 28 SNPs associated with LVESVi—robustly predicted DCM (HR 1.58 per SD increase in the polygenic score, $P = 6.4 \times 10^{-18}$ by Cox regression). That the common genetic variants in this polygenic score were identified after excluding individuals with known cardiac disease, and yet still predicted cardiomyopathy, suggests an intrinsic connection between the common genetic determinants of normal and pathologic variation in cardiac structure and function. Specifically, these results raise the possibility that, in some individuals, DCM may reflect the extreme of phenotypic variation, with important contributions from a high burden of common variants. This may, in part, explain the incomplete yield of genetic testing for rare variants in DCM. Future studies are required to determine the relative contributions of—and potential interplay between—common variants, rare variants, and environmental factors in the pathogenesis of DCM.

Second, genetic analyses of quantitative cardiac imaging traits may improve our understanding of the common genetic basis of cardiomyopathies. In our prior genetic analysis of the UK Biobank, we refined a heterogeneous heart failure phenotype to a specific, nonischemic cardiomyopathy subset, enabling detection of two DCM risk loci (near *BAG3* and *CLCNKA*) that associated with subclinical changes in LV structure and function[32]. Similarly, seminal GWAS of DCM yielded nine risk loci, but were limited by the recruitment of cardiomyopathy cases[7–9]. A collaborative effort to boost power by aggregating all heart failure subtypes across a number of studies yielded several common genetic loci for heart failure risk factors, i.e., coronary artery disease and atrial fibrillation, but few loci for cardiomyopathies[33]. By comparison, recent work analyzing the first 17,000 cardiac MRI studies from the UK Biobank reidentified eight loci previously found to be associated with cardiomyopathy or cardiac imaging traits[34]. Here, we pursued a genetic analysis, including a larger set of cardiac MRI studies from the UK Biobank (total sample size = 36,041). We reidentified all five loci harboring Mendelian cardiomyopathy-linked genes found in prior common variant analyses of DCM (*ALPK3*, *BAG3*, *FLNC*, *PLEKHM2*, and *TTN*). In addition, we discovered common genetic variants in loci near an additional 12 Mendelian cardiomyopathy-linked genes (*ACTN2*, *CSRP3*, *GATA4*, *MYH6*, *MYH7*, *NKX2–5*, *PLN*, *PTPN11*, *RBM20*, *RYR2*, *SHOC2*, and *TMEM43*, Supplementary Fig. 6). Finally, our PheWAS confirmed in an unbiased fashion that genetic loci linked to normal variation in cardiac MRI phenotypes were strongly associated with cardiomyopathies. Further analyses of MRI-derived cardiac traits may permit efficient study of the genetic determinants of cardiac structure and function in both health and disease, complementing the growing case–control genetic studies of cardiomyopathies.

Third, our results provide new insights into the role that common genetic variants play in determining the structure and function of the heart, even in the context of rare, high-impact mutations in cardiomyopathy-related genes. In a prior analysis of TTNtv, TTNtv carrier status associated with changes in LVEDV (+11.8 mL), LVESV (+7.7 mL), and LVEF (−2.8%) in individuals without clinical DCM[6]. In the present study, among carriers of cardiac-relevant TTNtv (those spliced into at least 90% of transcripts in the heart), a 1-SD increase in the LVESVi polygenic

score associated with comparable changes in LVEDV (+7.9 mL), LVESV (+7.2 mL), and LVEF (−2.6%). These findings emphasize the potential scope and impact of common genetic variants on cardiac structure and function, and suggest that the penetrance of high-impact rare variants may be influenced by carriers' polygenic backgrounds: for example, individuals with a cardiomyopathy-linked rare variant—but a favorable background of common genetic variants—may be less likely to develop a reduced LVEF. Future studies will be required to confirm these observations, and to ascertain whether these individuals might be protected from cardiomyopathy.

Finally, our results suggest that the costamere may play an important role in determining both normal and pathologic variation in myocardial traits. Costameres are cytoskeletal assemblies that connect the sarcomere—the basic structural unit of muscle—to the cell membrane and the extracellular matrix. Costameres include protein complexes such as the dystrophin glycoprotein complex (DGC) and the vinculin–talin–integrin system[35]. Both the DGC and the vinculin–talin–integrin system play roles in mechanical anchoring, while the vinculin–talin–integrin system also plays a role in mechanosensation and signal transduction[36]. Notably, the genes closest to four lead SNPs (FLNC, PTK2, PXN, and SSPN) play key roles in costamere biology, and two (SSPN and PXN) are also the most strongly prioritized genes at their respective loci by TWAS. Y-filamin (produced by FLNC) links the membrane-bound DGC with sarcomeric actin filaments[37]. PTK2 (also known as FAK) and PXN produce proteins (focal adhesion kinase and paxillin, respectively) that bind to vinculin and integrin to form part of the vinculin–talin–integrin system[38]. Finally, sarcospan, the protein product of SSPN, is a part of the DGC and is required for DGC–integrin interactions[39,40]. The emergence of these loci from our unbiased genetic analysis of cardiac MRI phenotypes points to the importance of the costamere within the cardiomyocyte, and suggests that compromise of its protein assemblies may contribute to the development of cardiomyopathies.

There are several limitations to our findings. First, our analyses were limited to older individuals of predominantly European ancestry, which may limit their applicability to younger individuals and those of other ancestries. Second, the cardiac measurements are derived from automated readings. Third, because the UK Biobank relied on hospitalization or death to assign disease status, unrecognized disease at baseline may have occurred for individuals without any pre-enrollment hospitalizations. Fourth, the commonly used GWAS P value significance threshold of $5 \times 10^{-8}$ in European populations was derived from Encyclopedia of DNA Elements (ENCODE) data in a small sample[41]; while this threshold remains commonly used, there is no universally agreed-upon P value threshold that accounts for larger sample sizes and rarer minor allele frequencies, although more stringent thresholds have been proposed[42,43].

In conclusion, we uncover common variants at 45 genetic loci not previously associated with left ventricular structure and function as measured by cardiac MRI, and reveal a robust link to DCM.

## Methods

**Study participants**. The UK Biobank is a richly phenotyped, prospective, population-based cohort[19]. In total, we analyzed 487,283 participants with genetic data who had not withdrawn consent as of October 2018. Analysis of the UK Biobank data was approved by the Partners Health Care institutional review board (protocol 2013P001840). Phenotypic analysis and genome sequencing of MESA participants in TOPMed was previously approved by the MESA field center institutional review boards (Columbia University, Johns Hopkins University, Northwestern University, University of Minnesota, University of California, Los Angeles, and Wake Forest University); all participants provided written consent[25].

**Phenotype refinement**. Disease phenotypes in the UK Biobank were defined using a combination of self-reported data (confirmed by a healthcare professional), hospital admission data, and death registry data. We defined nonischemic DCM as a billing code diagnosis of DCM (ICD-10 code I42.0) in the absence of coronary artery disease. Algorithms for identifying individuals with DCM, heart failure from any cause, and coronary artery disease in the UK Biobank are detailed in Supplementary Data File 1[32].

**Cardiac MRI measurements**. Cardiac MRI was performed with 1.5-Tesla scanners (MAGNETOM Aera, Syngo Platform vD13A, Siemens Healthcare) with electrocardiographic gating for cardiac synchronization[21]. Cardiac assessment was performed from the combination of several cine series using balanced steady-state free precession acquisitions, with post processing by cvi42 Version 5.1.1[22]. All measurements were provided by the UK Biobank. Because of known bias in the vD13A automated measurements, a bias correction was applied for LVEDV and LVESV measurements, using linear corrections derived from a UK cohort undergoing imaging on the same MRI platform[44].

Values for LVEF and SV were calculated from the LVEDV and LVESV using Eqs. (1) and (2).

$$LVEF = (LVEDV - LVESV) * (LVEDV)^{-1} \qquad (1)$$

$$SV = LVEDV - LVESV \qquad (2)$$

BSA-indexed values were produced for LVEDVi, LVESVi, and SVi by, respectively, dividing the values for LVEDV, LVESV, and SV by the individual's Mosteller BSA using Eq. (3)[45].

$$((Height\ in\ centimeters) * (weight\ in\ kilograms) * 3600^{-1})^{1/2} \qquad (3)$$

**Cardiac MRI sample selection and quality control**. We identified 39,298 individuals with cardiac MRI data. To account for errors in the automated volume measurement system, two cardiologists (JPP and KGA) manually reviewed images for samples having LVEDV or LVESV beyond 1.5 interquartile range below the first quartile or above the third quartile[46]. In total, 415 samples identified as having gross anatomic mistracings were rejected. In total, 1605 samples that did not pass genotyping quality control, detailed below, were excluded. Data from 1237 individuals with incident or prevalent heart failure, DCM, hypertrophic cardiomyopathy, or coronary artery disease prior to the date of cardiac MRI were excluded. After these exclusions, 36,041 samples remained for analysis (flow diagram displayed in Supplementary Fig. 1).

**Genotyping and quality control**. UK Biobank samples were genotyped on either the UK BiLEVE or UK Biobank Axiom arrays, then imputed into the Haplotype Reference Consortium panel and the UK10K + 1000 Genomes panel[18].

We performed genotyping quality control by excluding genotyped variants with call rate <0.95, imputed variants with INFO score <0.3, imputed or hard-called variants with minor allele frequency less than 0.001 in the overall UK Biobank population, or imputed or hard-called variants with an effective minor allele count of less than 100 in the subset of individuals with cardiac MRI data. To calculate the effective minor allele count, the minor allele frequency was multiplied by the number of alleles in the analysis, and by the imputation INFO score provided by the UK Biobank[18]. After exclusions, 13,660,711 autosomal variants were analyzed. Variant positions were keyed to GRCh37/hg19.

We performed sample quality control by excluding samples that had no imputed genetic data, a genotyping call rate < 0.98, a mismatch between submitted and inferred sex, sex chromosome aneuploidy (neither XX nor XY), exclusion from kinship inference, excessive third-degree relatives, or that were outliers in heterozygosity or genotype missingness rates as defined centrally by the UK Biobank[18].

**Heritability analysis**. Heritability attributable to single-nucleotide polymorphisms (SNPs) at the 784,256 directly genotyped sites that passed quality control was computed with the --reml option from BOLT-LMM (version 2.3.2)[47]. Cross-trait genetic correlations between LVESV, LVEDV, and LVEF were computed with the same command.

**Genome-wide association study**. We performed GWAS using linear mixed models with BOLT-LMM (version 2.3.2) to account for ancestral heterogeneity, cryptic population structure, and sample relatedness[47,48]. As BOLT-LMM requires a linkage disequilibrium panel, we used the European linkage disequilibrium panel provided with BOLT. We conducted a GWAS for the rank-based inverse normal-transformed values of each of seven cardiac MRI phenotypes: LVEDV, LVESV, LVEF, SV, LVEDVi, LVESVi, and SVi. Each GWAS was adjusted for the first five principal components of ancestry, sex, year of birth, age at the time of MRI, and the MRI scanner's unique identifier to account for batch effects. Variants with BOLT-LMM $P < 5 \times 10^{-8}$ were considered to be genome-wide significant.

LD score regression was performed with ldsc (version 1.0.0) to partition the genomic control factor lambda into polygenic and inflation components using the software's default settings[49], which include removing SNPs with MAF < 0.01,

indels, strand-ambiguous SNPs, and SNPs within the MHC region[50]. Allelic heterogeneity was assessed by clumping at each genome-wide significant locus to identify additional variants having trait association $P < 5 \times 10^{-8}$ and $r^2 < 0.2$ with other independent genome-wide significant variants within 500 kbs of the top variant at each locus using FUMA[51]. The most strongly associated SNP at each locus is referred to as the lead SNP. Lead SNPs were tested for deviation from HWE using the exact test with a HWE P threshold $<1 \times 10^{-6}$ [52].

**External validation using cardiac MRI traits in MESA**. We identified 2338 participants from MESA with whole-genome sequencing and cardiac MRI data available from MESA Exam 5[24–26]. We excluded 67 with a history of coronary artery disease or heart failure, and 87 whose cardiac MRI revealed evidence of myocardial scar with late gadolinium uptake, leaving 2184 for replication analysis; details of the cardiac MRI and whole-genome sequencing methods used in TOPMed/MESA are available in the Supplementary Methods. Of the 138 trait-SNP pairs (accounting for SNPs associated with multiple traits), 119 were identified within MESA via whole-genome sequencing. Using the *lme4* package in R, we performed regression with linear mixed models to test the additive genetic dosage of the SNP on each trait, adjusting for covariates, including sex, the age and age[2] at baseline visit, age and age[2] at the time of MRI, duration between the first and current visit, the first five principal components of ancestry, and the visit site. We performed a joint analysis on all 2184 participants, treating ancestry as a random effect. Six SNPs were removed from analysis because the linear mixed model did not converge due to boundary conditions.

We then aligned the effect alleles between the UK Biobank and the MESA results, and determined whether the effect direction was the same (both signs positive or both signs negative). We performed binomial tests comparing the number of SNPs in agreement to the total number of SNPs tested. Two null hypotheses were tested: (1) that each site had a 50% chance of having effects in the same direction, regardless of the P value, and (2) that each site had a 5% chance of having agreement in effect direction with nominal significance at one-tailed binomial $P < 0.05$.

**External validation using TTE traits in BioBank Japan**. Summary statistics made available by BioBank Japan from GWAS of three traits derived from TTE, which were similar to traits in our study: fractional shortening (similar to LVEF), left ventricular end-diastolic diameter (similar to LVEDV), and left ventricular end-systolic diameter (similar to LVESV)[14].

For those three paired traits, we aligned the effect alleles between the UK Biobank results and the BioBank Japan results, and determined whether the effect direction was the same (both signs positive or both signs negative). We determined the significance of the agreement between the two studies compared with chance with a two-tailed binomial test, testing against a null probability of agreement of 50% at each variant.

**Transcriptome-wide gene mapping**. TWAS identify genes whose expression is linked to phenotypes, integrating information about genetic associations with transcriptional variation and complex traits. This provides a complementary gene mapping and prioritization strategy in addition to the strategy of identifying the nearest gene to each GWAS lead SNP. For each of the seven cardiac MRI phenotypes, we performed a TWAS to identify the most strongly associated gene at each locus based on imputed *cis*-regulated gene expression[27,53–55]. We used the software package *FUSION* with eQTL data sourced from the GTEx Portal v7. Precomputed transcript expression reference weights for the left ventricle ($N = 5081$ genes) and the right atrial appendage ($N = 5670$ genes) were downloaded from the *FUSION* authors' website (see Data availability)[27,28]. Because the *FUSION* authors recommend applying the *ldsc* data preprocessing steps to GWAS summary statistics, we used the same input as was used for *ldsc*, described above, with the default settings applied. We ranked the genes within 500 kb of each lead SNP, consistent with the *FUSION* authors' approach[27].

**Tissue enrichment**. From the associated SNPs for each genotype, we evaluated the associations of 10,678 gene sets from MSigDB v6.2 and gene expression sets from GTEx using MAGMA[28,56,57]. Tissue enrichment tests were executed on the FUMA platform v1.3.4b[51]. A Bonferroni-corrected threshold of $\frac{0.05}{53 \text{ tissue types}} = 9.4 \times 10^{-4}$ was considered statistically significant enrichment of a tissue type.

**Mendelian cardiomyopathy gene set enrichment**. To create an unbiased list of genes associated with Mendelian cardiomyopathies and congenital heart abnormalities, we assembled genes from three commercially available cardiomyopathy gene panels (GeneDx Cardiomyopathy Panel, 207 Perry Parkway, Gaithersburg, MD, USA; Invitae Cardiomyopathy Comprehensive Panel, 1400 16th Street, San Francisco, CA, USA; Partners Laboratory for Molecular Medicine Pan Cardiomyopathy Panel, 77 Avenue Louis Pasteur Suite 250, Boston, MA, USA). Together, these panels contained 129 genes (Supplementary Table 8), all of which mapped to transcripts with HG19 coordinates. SNPsnap was used to generate 10,000 sets of SNPs that match the lead SNPs from the GWAS based on minor allele frequency, number of SNPs in linkage disequilibrium, distance to the nearest

gene, and gene density at the locus[58]. A SNP was considered to be near a Mendelian locus if it was within a radius of 500 kb upstream or downstream of any gene on the panel, a radius chosen to be the same as that used in the TWAS. Significance was assessed by permutation testing across the 10,000 SNP sets to determine the neutral expectation for the number of overlapping genes in loci with characteristics similar to ours, yielding a one-tailed P value.

**Polygenic score creation**. For each individual, we calculated a polygenic score by taking the beta for each lead SNP multiplied by the genetic dosage of the effect allele, and summing this value for each lead SNP for that trait from Supplementary Data File 3. We repeated this procedure for each of the seven traits, producing seven polygenic scores.

**PheWAS**. We performed a PheWAS in the 449,027 individuals with genetic data who had not undergone cardiac MRI. In total, 1591 ICD-10-derived PheCode phenotypes were available[30]. We tested the polygenic scores for association with the 1216 PheCode phenotypes that were present in 200 or more individuals in the UK Biobank, a case threshold that has previously been applied to single-variant PheWAS studies[59]. In addition, we tested the polygenic scores produced from each of the 7 cardiac traits for association with 96 manually curated disease phenotypes. Associations between the polygenic score and each phenotype were modeled with logistic regression, accounting for age at enrollment, sex, the genotyping array, and the first five principal components of ancestry as covariates. Among the manually curated disease phenotypes, Bonferroni correction for multiple testing (7 scores, 96 phenotypes) yielded a threshold for significance of logistic regression $P < 7.4 \times 10^{-5}$.

**Testing polygenic score association with incident DCM**. We assessed the relationship between the polygenic scores and DCM in the 362,922 participants who had not undergone cardiac MRI, were free from CHF, DCM, and CAD at baseline, and who were not identified by the UK Biobank as having third-degree or closer relatedness to another participant[18]. This sample included 380 individuals with incident DCM. We tested each score separately for association with incident DCM using a Cox proportional hazard model. This model was adjusted for sex, genotyping array, the first five principal components of ancestry, and the cubic basis spline of age at enrollment. Cox modeling was performed with the *survival* package in R 3.6.1 (R Foundation for Statistical Computing, Vienna, Austria). We defined statistical significance as a two-tailed Cox regression $P < (0.05/7 \text{ pheno-types}) = 0.007$.

For plotting, the samples were divided into deciles by LVESVi polygenic score. DCM incidence over time was plotted for the individuals in the lowest 10% of the LVESVi polygenic score, those in the highest 10%, and those in the middle 80% using the *survminer* package in R.

**Testing the polygenic score within TTNtv carriers**. Samples from the UK Biobank were prioritized for exome sequencing based on the presence of MRI data and linked hospital records[4]. Exomes were generated by the Regeneron Genetics Center, and reprocessed centrally by the UK Biobank according to Functional Equivalent standards[60]. Exomes were captured with the IDT xGen Exome Research Panel v1.0, targeting 39 million basepairs of the human genome. Sequencing was performed with 75 basepair paired-end reads on the Illumina NovaSeq 6000 platform using S2 flowcells. Alignment to GRCh38 was performed centrally with BWA-mem to generate a CRAM file for each sample. Variant calling was performed with GATK 3.0. Variants were hard-filtered if the inbreeding coefficient was $<-0.03$, or if none of the following were true: the read depth was greater than or equal to 10, the genotype quality was greater than or equal to 20, or the allele balance was greater than or equal to 0.2. In total, 49,997 exomes were available. Variants in *TTN* were annotated with the Ensembl Variant Effect Predictor version 95 with the --pick-allele flag[61]. LOFTEE was used to identify high-confidence loss-of-function variants in *TTN* (TTNtv): namely, stop-gained, splice-site disrupting, and frameshift variants[62].

Of the samples with cardiac MRI that passed all of the disease exclusion and quality control measures described above, 12,448 also had exome sequencing. Of these, 59 samples had TTNtv in exons spliced into more than 90% of transcripts in the heart (hig PSI), which are likely to be relevant to cardiac phenotypes[5]. An additional 692 participants were identified as putative carriers of high-PSI TTNtv based on their imputed genotypes; however, 66 of these participants had exome-sequencing data, and zero of the 66 were confirmed to have TTNtv by sequencing (Supplementary Table 14). Therefore, we limited our analysis to the 59 participants with high-PSI TTNtv that were identified by exome sequencing.

We applied the polygenic score most predictive of DCM (derived from the LVESVi) to the TTNtv carriers, and assessed whether that score was associated with LVESV, LVEDV, and LVEF in those individuals in a linear regression model after adjustment for the first five principal components of ancestry, genotyping array, sex, and the cubic basis spline of age at enrollment.

**Reporting summary**. Further information on research design is available in the Nature Research Reporting Summary linked to this article.

## Data availability

The raw UK Biobank data are made available to researchers from universities and other research institutions with genuine research inquiries, following IRB and UK Biobank approval. Individual-level sequence data for TOPMed: MESA whole genomes are available through restricted access via the TOPMed dbGaP Exchange Area (accession number phs001416.v1.p1 [https://www.ncbi.nlm.nih.gov/projects/gap/cgi-bin/study.cgi?study_id=phs001416.v1.p1]). The full GWAS summary statistics from the main analysis for each of the seven traits are available for download from the Broad Institute Cardiovascular Disease Knowledge Portal under the 'Downloads' tab at http://www.broadcvdi.org/. The PheCode PheWAS and the TWAS results are available in Supplementary Data Table 5. Precomputed GTEx v7 expression reference weights used for TWAS are available at http://gusevlab.org/projects/fusion/. All other data are contained within the article and its supplementary information, or are available upon reasonable request to the corresponding author.

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

## Acknowledgements

This work was conducted using the UK Biobank under application 7089. Whole-genome sequencing for "NHLBI TOPMed: Multi-Ethnic Study of Atherosclerosis (MESA)" (phs001416.v1.p1) was performed at the Broad Institute (3U54HG003067-13S1). Centralized read mapping and genotype calling, along with variant-quality metrics and filtering, were provided by the TOPMed Informatics Research Center (3R01HL-117626-02S1, contract HHSN268201800002I). Phenotype harmonization, data management, sample-identity QC, and general study coordination were provided by the TOPMed Data Coordinating Center (3R01HL-120393-02S1, contract HHSN268201800001I). We gratefully acknowledge the studies and participants who provided biological samples and data for TOPMed. MESA is supported by the National Heart, Lung, and Blood Institute (NHLBI) and contracts HHSN268201500003I, N01-HC-95159, N01-HC-95160, N01-HC-95161, N01-HC-95162, N01-HC-95163, N01-HC-95164, N01-HC-95165, N01-HC-95166, N01-HC-95167, N01-HC-95168, N01-HC-95169, UL1-TR-000040, UL1-TR-001079, UL1-TR-001420, UL1-TR-001881, and DK063491. Whole-genome sequencing for the Trans-Omics in Precision Medicine (TOPMed) program was supported by the NHLBI. This work was funded by the National Institutes of Health RO1 HL127564 (to S.K.), the National Human Genome Research Institute of the US National Institutes of Health under award number 5UM1HG008895 (to S.K.), and the Ofer and Shelly Nemirovsky Research Scholar Award from Massachusetts General Hospital (to S.K.). K.G.A. is supported by the American Heart Association Institute for Precision Cardiovascular Medicine (17IFUNP33840012). J.P.P. is supported by the John S. LaDue Memorial Fellowship for Cardiovascular Research. S.A.L. is supported by NIH grant 1R01HL139731 and American Heart Association 18SFRN34250007. P.T.E. is supported by the Fondation Leducq (14CVD01), the NIH (1RO1HL092577, R01HL128914, and K24HL105780) and the American Heart Association (18SFRN34110082).

## Author contributions

J.P.P., S.K., and K.G.A. conceived the study. J.P.P., M.W., M.C., J.Y., and X.G. performed the computational analyses. J.P.P and K.G.A. wrote the paper. All authors contributed to the analysis plan or provided critical revisions.

## Competing interests

J.P.P. and A.B. have minor consulting roles with Maze Therapeutics. A.P. is a Venture Partner at GV, a venture capital group within Alphabet. In that capacity, he makes investments in companies in both the life sciences and data sciences, many of which are cardiovascular in focus. S.A.L. receives sponsored research support from Bristol Myers Squibb/Pfizer, Bayer AG, and Boehringer Ingelheim, and has consulted for Bristol Myers Squibb/Pfizer and Bayer AG. P.T.E. is supported by a grant from Bayer AG to the Broad Institute focused on the genetics and therapeutics of cardiovascular diseases, and has served on advisory boards or consulted for Bayer AG, Quest Diagnostics, and Novartis. S.K. is an employee of Verve Therapeutics, and holds equity in Verve Therapeutics, Maze Therapeutics, Catabasis, and San Therapeutics. He is a member of the scientific advisory boards for Regeneron Genetics Center and Corvidia Therapeutics; he has served as a consultant for Acceleron, Eli Lilly, Novartis, Merck, Novo Nordisk, Novo Ventures, Ionis, Alnylam, Aegerion, Haug Partners, Noble Insights, Leerink Partners, Bayer Healthcare, Illumina, Color Genomics, MedGenome, Quest, and Medscape; he reports patents related to a method of identifying and treating a person having a predisposition to or afflicted with cardiometabolic disease (20180010185) and a genetics risk predictor (20190017119). The remaining authors declare no competing interests.
