## [Peer Review File · Nature Communications]

Reviewers' Comments:

Reviewer #1:

Remarks to the Author:

This is a genome-wide association study (GWAS) of seven cardiac magnetic resonance imaging (MRI)-derived left ventricular measurements in 29,041 UK Biobank participants. The authors aimed to discover common variants that associate with cardiac structure and function and to assess if those variants confer risk of developing dilated cardiomyopathy (DCM). They found 37 distinct loci that associated with one or more cardiac MRI phenotypes using the P value of 5×10^{-8} as threshold for genome-wide significance. Of the 37 loci, 26 have not been described before in relation with DCM or cardiac imaging phenotypes. This is a classic common variant GWAS, using a large and well phenotyped public cohort, performed by an experienced group. The paper is clear and well written. I have a few comments.

1. The selection of the significance threshold of $P < 5 \times 10^{-8}$ needs to be rationalized. The commonly used GWAS threshold of 5×10^{-8} is based on estimation of the multiple testing burden from testing only common variants ($MAF > 5\%$) for association in a European sample of 1000 cases and 1000 thousand controls (estimated to correspond to one million independent tests) (Pe'er Genetic Epidemiology 2008). This multiple test burden increases with sample size and by testing lower frequency variants.
2. Given a significance threshold of $P < 5 \times 10^{-8}$ many of the claimed associations are borderline significant and no form of replication is attempted. This should be addressed.
3. The authors state that five loci previously reported for echocardiographic measures are GWS in the current study and that they „recovered“ six of nine loci previously identified in studies of heart failure / cardiomyopathy. The term recovered is unclear, please be more specific. It would be of improvement to the paper to provide a supplementary table with association results for previously reported common variants for cardiac imaging, heart failure and cardiomyopathy. So for example, the five echo loci that were not GWS in the current study, did they associate at all, replicate?
4. Please show in Table 1 which loci have been reported before for cardiac imaging, heart failure or cardiomyopathy.
5. The following description in the results section is quite unclear, please attempt to improve: „Out of 100,000 simulations, none was linked to more than 14 Mendelian genes (Supplemental Figure 6a); in contrast, our loci are near 17 Mendelian genes from the gene panel (Supplemental Table 5), were significantly enriched for Mendelian cardiomyopathy genes ($P < 1 \times 10^{-5}$).“
6. Please provide an index for the supplementary file.

Reviewer #2:

Remarks to the Author:

The authors report a GWAS of left ventricular geometry and function in 29,000 individuals in UK Biobank, the findings from which they use to examine the relationship between genetic determinants of left ventricular phenotypes and risk of dilated cardiomyopathy (DCM). They identify 37 loci associated with at least one of seven phenotypes of which 26 appear not to have been previously reported. In post-GWAS analysis, the authors report that the loci identified in the GWAS are enriched for genes expressed in the myocardium and are, through proximity, enriched for genes containing known cardiomyopathy-causing mutations. A polygenic risk score for LVESVi associated with risk of DCM in participants for whom MRI data were unavailable, and appeared to modulate LVESVi in participants with TTN truncating variants identified through whole exome sequencing.

The selection and handling of phenotypes and GWAS analysis are appropriately conducted and described, and on the whole the post-GWAS analysis is reasonable. I have a number of comments on the manuscript and the conduct of the analysis it describes (page numbers refer to the pages of the PDF).

Firstly, and most significantly, the manuscript is most notable for the absence of formal signal-to-gene mapping. The authors describe using both straightforward proximity-based mapping whereby signals are assigned to their closest gene, or mapping based on the closest known cardiomyopathy-causing gene. Neither of these approaches is very rigorous and both leave open the possibility of mis-mapping. A number of the strands of analysis conducted and conclusions drawn rely on the mapping of GWAS signals to genes and the lack of rigour in this process risks casting doubt on those subsequent findings. The authors should seriously consider adding formal signal-to-gene mapping analysis to the manuscript using, for example, colocalization with gene expression in the publicly available GTEx myocardial tissues. It may transpire that proximity-based mapping will yield the best possible mapping at some loci, but to make this assumption across all results without testing it risks undermining much of the downstream analysis.

Page 5 ll.85-86 - "Nine loci with variants that distinguish cases from controls...": "Distinguish" suggests the variants perfectly discriminate between cases and controls, which is unlikely to be the case; these variants increase risk of disease rather than being both necessary and sufficient. The language here should be amended accordingly.

Page 8 l.154 - The authors should state whether the variants near PLN, SH2B3/ATXN2, MTSS1, SMARCB1 and CDKN1A were associated with the same traits in this analysis as they were in the Wild et al. and Kanai et al. analyses? If not, the inconsistencies should be discussed.

Page 9 l. 180 - I am concerned at the low threshold for case numbers (n=20) used for inclusion of traits in the GWAS, which inevitably impairs the power of the analysis to detect true positive signals particularly given the large number of tests undertaken within the pheWAS analysis. The authors should justify the use of such a low threshold and should provide a power calculation to demonstrate the power of this analysis once multiple testing has been accounted for.

Page 9 ll.186-188, supplementary table 6 and supplementary figure 7 - The authors should comment on the presence of several PRS associations with thyroid disease in supplementary table 6. Given the known relationship between thyrotoxicosis and heart failure this is likely pertinent to the manuscript. Also, the authors should explain the rationale for showing only associations with $p < 10^{-5}$, and whether multiple testing was accounted for in the pheWAS.

Page 11 l.220 - Participants with TTN truncating variants were identified using WES data alone. Were any known TTNTvs represented on the UKB GWAS arrays and, if so, could these have been used to identify additional individuals beyond those for whom WES was available? Furthermore, the authors should justify their decision to report findings in a sample of just 53 individuals where power to detect the effects of common variants is likely to be limited and false positive findings are more likely than in a larger sample.

Page 14 ll. 282-297 - As noted above, without formal mapping of lead SNPs to likely causal genes the claims in this paragraph seem overstated. While it is possible, or even likely, that the lead SNPs act through FLNC, SSPN and PXN the authors have not sought to confirm this and the wording should be moderated to be more circumspect.

“Analysis of cardiac magnetic resonance imaging traits in 36,000 individuals reveals shared genetic basis with dilated cardiomyopathy”

Reviewer #1

1. The selection of the significance threshold of $P < 5 \times 10^{-8}$ needs to be rationalized. The commonly used GWAS threshold of 5×10^{-8} is based on estimation of the multiple testing burden from testing only common variants (MAF > 5%) for association in a European sample of 1000 cases and 1000 thousand controls (estimated to correspond to one million independent tests) (Pe'er Genetic Epidemiology 2008). This multiple test burden increases with sample size and by testing lower frequency variants.

Author response:

We agree with the Reviewer that the threshold of $P < 5 \times 10^{-8}$ is imperfect for assessing genome-wide significance, particularly as sample sizes have increased and the MAF of assessed variants has decreased since the Pe'er 2008 article was published. In our current study, we identified 138 SNP-trait associations significant at $P < 5 \times 10^{-8}$, 57 of which represent distinct loci after accounting for the same loci being identified in two or more traits. Empirically, if we relaxed our P threshold for significance to 5×10^{-7} , we would identify 257 significant SNP-trait associations (109 distinct), whereas if we made it more stringent at 5×10^{-9} , we would identify 91 significant SNP-trait associations (36 distinct). Since no alternative P value threshold has become widely agreed upon in the field, we continue to use $P < 5 \times 10^{-8}$ as our genome-wide significance threshold. However, in light of this (and at the Reviewer's suggestion), we have placed additional emphasis on replication of our primary findings, as discussed in the next section.

Manuscript change:

In the Study limitations section, we have now added the following text:

“Fourth, the standard GWAS P value significance threshold of 5×10^{-8} in European populations was derived from Encyclopedia of DNA Elements (ENCODE) data in a small sample (Pe'er et al., 2008); while this threshold remains commonly used, there is no universally agreed upon P value threshold that accounts for larger sample sizes and rarer minor allele frequencies.”

2. Given a significance threshold of $P < 5 \times 10^{-8}$ many of the claimed associations are borderline significant and no form of replication is attempted. This should be addressed.

Author response:

We appreciate this important suggestion by the Reviewer and have now pursued replication of our primary findings in two independent cohorts with genetic and cardiac imaging data:

- a) External validation in the Multiethnic Study of Atherosclerosis (MESA). We pursued external validation of the lead SNPs from our primary analysis using genetic and cardiac MRI data from over 2,000 participants in MESA. Of the 138 lead SNP-trait associations (across 7 MRI traits) with available whole-genome sequencing data in MESA, 113 could be identified and statistically modeled. 99 of the 113 SNP-trait associations had effects in the same direction in the UK Biobank and in MESA (87.6% concordance; binomial test with two-tailed $P = 6.1 \times 10^{-17}$, given a chance expectation of 50% at each of the 113 sites). 27 SNP-trait associations reached nominal significance ($P < 0.05$), of which, 26 had directionally consistent effects in MESA and the UK Biobank (binomial test with two-tailed $P = 5.4 \times 10^{-11}$, given a chance expectation of 5% at each of 113 sites).
- b) External validation in BioBank Japan. We tested the association of lead SNPs with analogous transthoracic echocardiography (TTE)-derived traits of LV structure and function using summary statistics from genetic association studies of TTE in BioBank Japan. In BioBank Japan, of 46 SNP-trait associations (across 3 analogous TTE traits) with available genetic information, 39 had effects in the same direction as in UK Biobank (84.8% agreement; binomial test with two-tailed $P = 1.8 \times 10^{-6}$).

Manuscript changes:

(a) External validation with cardiac MRI data from MESA. Our approach for external validation with cardiac MRI data from MESA is now described in the **Methods** section:

“We identified 2,338 participants from MESA with whole genome sequencing and cardiac MRI data available from MESA Exam 5 (Bluemke et al., 2008; Natarajan et al., 2018; Yoneyama et al., 2017). We excluded 67 with a history of coronary artery disease or heart failure, and 87 whose cardiac MRI revealed evidence of myocardial scar with late gadolinium uptake, leaving 2,184 for replication analysis; details of the cardiac MRI and whole genome sequencing methods used in TOPMed/MESA are available in the Supplemental Methods. Of the 138 trait-SNP pairs (accounting for SNPs associated with multiple traits), 119 were identified within MESA via whole genome sequencing. Using the lme4 package in R, we performed regression with linear mixed models to test the additive genetic dosage of the SNP on each trait, adjusting for covariates including sex, the age and age² at baseline visit, age and age² at the time of MRI, duration between first and current visit, the first five principal components of ancestry, and the visit site. We performed a joint analysis on all 2,184 participants, treating ancestry as a random effect. Six SNPs were removed from analysis because the linear mixed model did not converge due to boundary conditions.

We then aligned the effect alleles between the UK Biobank results and the MESA results and determined whether the effect direction was the same (both signs positive or both signs negative). We performed binomial tests comparing the number of SNPs in

agreement to the total number of SNPs tested. Two null hypotheses were tested: (1) that each site had a 50% chance of having effects in the same direction regardless of P value, and (2) that each site had a 5% chance of having agreement in effect direction with nominal significance at $P < 0.05$.”

The **Supplemental Methods** include the following:

“MESA cardiac magnetic resonance imaging

MESA is a longitudinal cohort study of cardiovascular disease in 6,814 participants aged 44-84 years at enrollment and recruited in six US sites. A baseline examination was held from 2000-2002 and follow up exams from 2002 – 2012. Cardiovascular MRI was performed on 5,004 participants at baseline and 3,015 during a follow-up examination (“Exam 5”). The MRI protocol during Exam 5 used a similar technique (steady-state free precession) to that used in the UK Biobank, and was therefore chosen for comparison against the same 7 traits in the main analysis.”

“MESA whole genome sequencing

Whole genome sequencing and quality control of the MESA participants was described previously². In brief, ~4,610 MESA participants were chosen for whole genome sequencing at the Broad Institute (Cambridge, MA), with written consent and IRB approval (IRB #2016P001308). Sequencing was performed on Illumina HiSeqX platform with 151-bp paired-end reads. Samples were excluded from downstream variant calling if the estimated contamination was $\geq 3\%$ or $<95\%$ of the genome was covered to at least 10x read depth. Variants were called with GATK version 3 HaplotypeCaller³.”

Findings from our external validation strategy with cardiac MRI data from MESA are now summarized in the **Results** section:

“In MESA, we studied 2,184 participants with whole-genome sequencing and cardiac MRI data available, and who did not have cardiovascular disease or late gadolinium enhancement consistent with myocardial scar (**Supplemental Table 5**) (Bluemke et al., 2008; Natarajan et al., 2018; Yoneyama et al., 2017). Of the 138 genome-wide significant SNP-trait associations in our discovery analysis (accounting for SNPs significantly associated with multiple traits), 113 were also available in MESA and able to be statistically modeled. 99 of 113 SNP-trait associations had effects in the same direction for UK Biobank and MESA (87.6% concordance; binomial test with two-tailed $P = 6.1 \times 10^{-17}$, given a chance expectation of 50% at each of 113 sites). Of the 27 SNP-trait associations with a P value < 0.05 , 26 had an effect in the same direction as in the UK Biobank (binomial test with two-tailed $P = 5.4 \times 10^{-11}$, given a chance expectation of 5% at each of 113 sites). Validation results for each SNP are available in **Supplemental Table 6**.”

(b) External validation with transthoracic echocardiography (TTE) data in BioBank Japan. Our approach for external validation in BioBank Japan is now included in the **Methods** section:

“Summary statistics made available by BioBank Japan from GWAS of three traits derived from TTE which were similar to traits in our study: fractional shortening (similar to LVEF), left ventricular end diastolic diameter (similar to LVEDV), and left ventricular end systolic diameter (similar to LVESV) (Kanai et al., 2018).

For those 3 paired traits, we aligned the effect alleles between the UK Biobank results and the BioBank Japan results and determined whether the effect direction was the same (both signs positive or both signs negative). We determined the significance of the agreement between the two studies compared to chance with a two-tailed binomial test, testing against a null probability of agreement of 50% at each variant.”

Findings from our external validation with TTE are now summarized in the **Results** section:

“We then performed cross-modality validation using summary statistics from a genome-wide association study in BioBank Japan of 19,000 participants of Japanese ancestry with TTE data (Kanai et al., 2018), including three traits analogous to LVEDV, LVESV, and LVEF. We were able to identify data from BioBank Japan for 46 SNP-trait associations (out of 76 total across the three traits, accounting for sites associated with multiple traits). Of these 46 SNP-trait associations, 39 had an effect in the same direction as in the UK Biobank (84.8% agreement; binomial test with two-tailed $P = 1.8 \times 10^{-6}$; **Supplemental Table 6**).”

We have also included **Supplemental Table 5** and **Supplemental Table 6** at the end of this document.

3. The authors state that five loci previously reported for echocardiographic measures are GWS in the current study and that they „recovered“ six of nine loci previously identified in studies of heart failure / cardiomyopathy. The term recovered is unclear, please be more specific. It would be of improvement to the paper to provide a supplementary table with association results for previously reported common variants for cardiac imaging, heart failure and cardiomyopathy. So for example, the five echo loci that were not GWS in the current study, did they associate at all, replicate?

Author response:

We thank the reviewer for noting this point of confusion. We have revised this sentence to clarify the text. Also, we appreciate the suggestion to include a summary of previously identified loci/variants and their associations within the UK Biobank; we have therefore compiled such a table and included it in the manuscript.

Manuscript change:

In the **Results** section, we have removed the word “recovered” and adjusted our language to clarify this point:

“Six of the nine SNPs previously identified at exome- or genome-wide significance in common variant studies of cardiomyopathy were also associated at genome-wide significance with at least one cardiac MRI phenotype in our study, including SNPs near *ALPK3*, *BAG3*, *CLCNKA/HSPB7*, *FHOD3*, *FLNC*, and *TTN* (Esslinger et al., 2017; Meder et al., 2014; Villard et al., 2011).”

To provide a resource reviewing previously reported common variants, we have added **Supplemental Table 4** and refer to this table within the results:

“Our data provide nominal supporting evidence ($P < 1 \times 10^{-3}$) for SNPs at two of the remaining three loci. The lead SNPs from these prior studies, as well as the association P value from the most strongly associated cardiac MRI phenotype from our study, are available in **Supplemental Table 4**.”

We have also included **Supplemental Table 4** at the end of this document.

4. Please show in Table 1 which loci have been reported before for cardiac imaging, heart failure or cardiomyopathy.

Author response:

We appreciate this helpful suggestion and have revised **Table 1** accordingly.

Manuscript change:

We have now added a column to **Table 1** to indicate which of the prior studies also identified each locus as genome-wide significant.

The updated **Table 1** is included at the end of this document.

5. The following description in the results section is quite unclear, please attempt to improve: „Out of 100,000 simulations, none was linked to more than 14 Mendelian genes (Supplemental Figure 6a); in contrast, our loci are near 17 Mendelian genes from the gene panel (Supplemental Table 5), were significantly enriched for Mendelian cardiomyopathy genes ($P < 1 \times 10^{-5}$).“

Author response:

We agree that this point may be simplified and have re-written the paragraph to more clearly communicate these results.

Please note that the number of Mendelian genes reported has increased, because the primary analysis now includes an additional 7,000 cardiac MRIs. We have also harmonized all distance cutoffs in the manuscript to reflect a 500kb radius, including the radius of a locus, the radius in which TWAS results are prioritized, and the radius in which Mendelian genes are sought.

Manuscript change:

We have now modified the relevant text to try to clarify this section of the **Results**:

“We then asked whether the GWAS loci that we identified for cardiac structure and function were more enriched for known cardiomyopathy genes than expected based on chance (**Supplemental Table 7**). We compared the likelihood of an overlap between known cardiomyopathy genes and our GWAS loci or matched control loci (please see **Methods** for details). In 10,000 simulations, we found a significant enrichment in Mendelian cardiomyopathy-linked genes at the cardiac MRI GWAS loci (one-tailed $P < 1 \times 10^{-4}$, **Supplemental Figure 6**).”

6. Please provide an index for the supplementary file.

Author response & Manuscript change:

We have now prefaced the supplement with an index.

Reviewer #2

Firstly, and most significantly, the manuscript is most notable for the absence of formal signal-to-gene mapping. The authors describe using both straightforward proximity-based mapping whereby signals are assigned to their closest gene, or mapping based on the closest known cardiomyopathy-causing gene. Neither of these approaches is very rigorous and both leave open the possibility of mis-mapping. A number of the strands of analysis conducted and conclusions drawn rely on the mapping of GWAS signals to genes and the lack of rigour in this process risks casting doubt on those subsequent findings. The authors should seriously consider adding formal signal-to-gene mapping analysis to the manuscript using, for example, colocalization with gene expression in the publicly available GTEx myocardial tissues. It may transpire that proximity-based mapping will yield the best possible mapping at some loci, but to make this assumption across all results without testing it risks undermining much of the downstream analysis.

Author response:

We appreciate the Reviewer's helpful suggestion. To address this point, we have now performed a transcriptome-wide association study (TWAS) using GTEx expression data from the left ventricle and the right atrial appendage, using the *FUSION* software. In addition to the manuscript changes that we detail below, we also anticipate making the TWAS results available for download in the **Data Tables**.

Manuscript change:

The **Methods** section has been updated with the following description:

“Transcriptome-wide association studies (TWAS) identify genes whose expression is linked to phenotypes, integrating information about genetic associations with transcriptional variation and complex traits. This provides a complementary gene mapping and prioritization strategy in addition to the strategy of identifying the nearest gene to each GWAS lead SNP. For each of the seven cardiac MRI phenotypes, we performed a TWAS to identify the most strongly associated gene at each locus based on imputed cis-regulated gene expression (Gamazon et al., 2015; Gusev et al., 2016, 2018; Zhu et al., 2016). We used the software package *FUSION* with eQTL data sourced from the GTEx Portal v7. Precomputed transcript expression reference weights for the left ventricle (N=5,081 genes) and the right atrial appendage (N=5,670 genes) were downloaded from the *FUSION* authors' website (see **Data Availability**) (Gusev et al., 2016; Lonsdale et al., 2013). Because the *FUSION* authors recommend applying the *ldsc* data preprocessing steps to GWAS summary statistics, we used the same input as was used for *ldsc*, described above, with the default settings applied. We ranked the genes within 500 kilobases of each lead SNP, consistent with the *FUSION* authors' approach (Gusev et al., 2016).”

The **Results** section has been updated as follows:

“We performed a transcriptome-wide association study (TWAS) to prioritize genes within 500kb of each locus based on expression in the left ventricle and right atrial appendage(Gusev et al., 2016; Lonsdale et al., 2013). TWAS-based prioritization is complementary to the approach of selecting the nearest gene at a locus(Gusev et al., 2016; Wainberg et al., 2019). The most strongly associated gene at each locus from the TWAS is annotated in **Table 1**. Seven of the cardiomyopathy-linked genes from **Supplemental Table 8** (*ACTN2*, *ALPK3*, *MYH6*, *NKX2-5*, *PLN*, *PTPN11*, and *SHOC2*) were the genes most strongly prioritized by TWAS at their respective loci for at least one trait. Three cardiomyopathy-linked genes (*BAG3*, *CSRP3*, and *TTN*) were not candidates for inclusion in the TWAS, because they were below the inclusion threshold based on the Genotype-Tissue Expression (GTEx) expression quantitative trait locus (eQTL) P value in both the left ventricle and the right atrial appendage.”

Table 1 has been updated to include the top TWAS-prioritized gene at each locus. **Table 1** and **Supplemental Table 8** are also included at the end of this document.

Page 5 II.85-86 - “Nine loci with variants that distinguish cases from controls...”:
“Distinguish” suggests the variants perfectly discriminate between cases and controls, which is unlikely to be the case; these variants increase risk of disease rather than being both necessary and sufficient. The language here should be amended accordingly.

Author response:

We thank the Reviewer for this helpful feedback and have revised the text to be more precise.

Manuscript change:

We have updated the Introduction to read:

“These case-control studies identified nine loci significantly associated with DCM, five of which contain genes that also harbor rare DCM-causing mutations (*TTN*, *ALPK3*, *BAG3*, *FLNC*, and *PLEKHM2*).”

Page 8 I.154 – The authors should state whether the variants near *PLN*, *SH2B3/ATXN2*, *MTSS1*, *SMARCB1* and *CDKN1A* were associated with the same traits in this analysis as they were in the Wild et al. and Kanai et al. analyses? If not, the inconsistencies should be discussed.

Author response:

We evaluated overlapping loci with the Wild and Kanai results based on analogous traits. For example, left ventricular internal diameter at end diastole (LVIDd) on TTE is analogous to left ventricular end diastolic volume (LVEDV) on MRI. We now specify those traits that are analogous between TTE and cardiac MRI, and stipulate that we evaluated such analogous

traits. The results within European-ancestry participants were consistent, while they were mixed in the Japanese-ancestry cohort.

Manuscript change:

We have updated the **Results** to read:

“Five of the 10 loci previously discovered through genetic analyses of echocardiographic measurements were among our genome-wide significant loci, including SNPs in loci near *PLN*, *SH2B3/ATXN2*, *MTSS1*, *SMARCB1/DERL3*, and *CDKN1A* (Kanai et al., 2018; Wild et al., 2017). The comparison to the echocardiographic studies was performed on a per-trait basis, considering fractional shortening on TTE to be analogous to LVEF on MRI, left ventricular internal diameter at end diastole (LVIDd) to be analogous to LVEDV, and left ventricular internal diameter at end systole (LVIDs) to be analogous to LVESV. Of the four loci previously identified in a GWAS of cardiac traits measured with TTE in participants of European ancestry (Wild et al., 2017), all four were also significantly associated with an analogous trait in our study (**Supplemental Table 4**). In comparison, two of six loci identified in a GWAS in participants of Japanese ancestry were significantly associated in our study (Kanai et al., 2018), a lower proportion that may reflect ancestry-specific patterns of linkage disequilibrium.”

We have also included **Supplemental Table 4** at the end of this document.

Page 9 l. 180 – I am concerned at the low threshold for case numbers (n=20) used for inclusion of traits in the GWAS, which inevitably impairs the power of the analysis to detect true positive signals particularly given the large number of tests undertaken within the pheWAS analysis. The authors should justify the use of such a low threshold and should provide a power calculation to demonstrate the power of this analysis once multiple testing has been accounted for.

Author response:

We agree that small case numbers limit statistical power to detect significant associations between the polygenic scores and diseases in the PheWAS, likely leading to many false negatives. Given this concern, we have now set a higher minimum case number of N=200.

This modification of minimum case number is based on the approach used by Verma, *et al*, “A simulation study investigating power estimates in phenome-wide association studies,” *BMC Bioinformatics* 2018. Their conclusions included the rule of thumb that a sample size of 200 cases or more may be effective for common variants. Therefore, we have updated the text and figures based on a case threshold of 200 samples (leading to 1,218 PheCode phenotypes being tested, from 1,591 previously).

Manuscript change:

We have now updated the **Methods** as follows:

We performed a PheWAS in the 449,027 individuals with genetic data who had not undergone cardiac MRI. We tested the polygenic scores for association with 1,591 ICD-10-derived PheCode phenotypes (Wu et al., 2019). 1,216 of the PheCode phenotypes were present in 200 or more individuals in the UK Biobank, a case threshold that has previously been applied to single-variant PheWAS studies (Verma et al., 2018). In addition, we tested the polygenic scores produced from each of the 7 cardiac traits for association with 96 manually curated disease phenotypes. Associations between the polygenic score and each phenotype were modeled with logistic regression, accounting for age at enrollment, sex, the genotyping array, and the first 5 principal components of ancestry as covariates. Among the manually curated disease phenotypes, Bonferroni correction for multiple testing (7 scores, 96 phenotypes) yielded a threshold for significance of $P < 7.4 \times 10^{-5}$.

We have now updated the **Results** as follows:

“We also produced polygenic scores for each trait, weighting the genetic dosage by the effect size of the lead SNPs from each GWAS from **Table 1**. We performed a cross-sectional phenome-wide association study (PheWAS) in 449,000 UK Biobank participants to assess the relationship between each of the seven polygenic scores and disease phenotypes. We first performed a PheWAS using a broad set of PheCodes (Wu et al., 2019), 1,216 of which were present in 200 or more participants. As anticipated, this analysis showed an enrichment for cardiac diseases. We then performed a PheWAS for 96 diseases using curated definitions (as defined in **Supplemental Table 1**). Among our curated disease traits, DCM emerged as the disease most strongly associated with the polygenic scores for LVEF, LVESV, and LVESVi. Among these, the LVESVi polygenic score had the single strongest relationship with DCM (OR 1.51 per standard deviation [SD] increase in LVESVi polygenic score; $P = 8.5 \times 10^{-34}$; **Figure 2**). The LVEDVi and SV polygenic scores had strong inverse relationships with hypothyroidism, an observation consistent with invasive studies that found reduced LVEDV and SV in hypothyroid patients, attributed to hemodynamic loading conditions (Wieshammer et al., 1988). The LVEDV and SV scores, both of which have contributions from the major histocompatibility complex (MHC), were most strongly inversely associated with psoriasis. Each of the seven polygenic scores had several significant associations with disease phenotypes, even after adjusting for multiple testing (Bonferroni-adjusted significance threshold $P = 5.9 \times 10^{-6}$ in the PheCode PheWAS, $P = 7.4 \times 10^{-5}$ in the manually curated disease phenotype PheWAS). The strongest relationships between each of the 7 polygenic scores and the disease phenotypes are detailed in **Supplemental Figure 7**. Manually curated disease phenotypes with Bonferroni-adjusted significant P values are displayed in **Supplemental Table 9**; the full table of PheWAS results for PheCodes is available for download (see **Data Availability**).”

We will be including the PheCode PheWAS results for samples with $N \geq 200$ in the downloadable **Data Table**. We have included **Table 1**, **Supplemental Table 9**, and **Figure 2** at the end of this document.

Page 9 ll.186-188, supplementary table 6 and supplementary figure 7 - The authors should comment on the presence of several PRS associations with thyroid disease in supplementary table 6. Given the known relationship between thyrotoxicosis and heart failure this is likely pertinent to the manuscript. Also, the authors should explain the rationale for showing only associations with $p < 10^{-5}$, and whether multiple testing was accounted for in the pheWAS.

Author response:

We appreciate the recommendation to address thyroid disease, and have amended the text accordingly.

Regarding the display of associations with $P < 10 \times 10^{-5}$ in the table containing manually curated disease phenotypes (previously Supplemental Table 6, now labeled **Supplemental Table 9**), we chose that threshold because it is the Bonferroni-adjusted P value threshold ($P < 0.05 / [96 \text{ traits} \times 7 \text{ scores}]$). We now clarify that in the manuscript text, which we have copied below.

Manuscript change:

We have now updated the **Results** to address thyroid disease as follows:

“The LVEDVi and SV polygenic scores had strong inverse relationships with hypothyroidism, an observation consistent with invasive studies that found reduced LVEDV and SV in hypothyroid patients, attributed to hemodynamic loading conditions (Wieshammer et al., 1988). The LVEDV and SV scores, both of which have contributions from the major histocompatibility complex (MHC), were most strongly inversely associated with psoriasis.”

We have now added the following text to the **Results** to provide guidance about which associations may be considered Bonferroni significant:

“Each of the seven polygenic scores had several significant associations with disease phenotypes, even after adjusting for multiple testing (Bonferroni-adjusted significance threshold $P = 5.9 \times 10^{-6}$ in the PheCode PheWAS, $P = 7.4 \times 10^{-5}$ in the manually curated disease phenotype PheWAS).”

Page 11 I.220 – Participants with TTN truncating variants were identified using WES data alone. Were any known TTNtv represented on the UKB GWAS arrays and, if so, could these have been used to identify additional individuals beyond those for whom WES was available? Furthermore, the authors should justify their decision to report findings in a sample of just 53 individuals where power to detect the effects of common variants is likely to be limited and false positive findings are more likely than in a larger sample.

Author response:

We appreciate the helpful suggestion to consider imputed TTNtv. To address this, we first attempted to assess the quality of TTNtv imputation. We found 66 participants (among 50,000 with exome sequencing) who had imputed genotypes indicating that they had a TTNtv. We then checked for the presence of this imputed TTNtv within their sequence data to confirm the TTNtv. However, we found that 0 of the imputed TTNtv were confirmed by sequencing; therefore, we did not include imputed TTNtv, and we only included directly sequenced TTNtv. We felt that this lack of sequencing confirmation of imputed TTNtv may be of interest to readers, and so it is now addressed briefly in the text.

We agree that 53 samples with TTNtv (now 59 due to the additional 7,000 samples with cardiac MRI data that became available during revision) is a small number, and that underpowered settings are enriched for false positives. However, to our knowledge, this represents the largest number of TTNtv carriers with cardiac MRI data that has been described to date. For example, Roberts *Sci Transl Med* 2015, described 42 TTNtv carriers with cardiac MRI data, while Schafer *Nat Genet* 2017 described 15 TTNtv carriers with cardiac MRI data.

Manuscript change:

In the **Methods**, we have now described our assessment of imputed TTNtv:

“An additional 692 participants were identified as putative carriers of high-PSI TTNtv based on their imputed genotypes; however, 66 of these participants had exome sequencing data, and zero of the 66 were confirmed to have TTNtv by sequencing (**Supplemental Table 13**). Therefore, we limited our analysis to the 59 participants with high-PSI TTNtv that were identified by exome sequencing.”

We have added more guarded language to the **Discussion**:

“Future studies are required to determine the relative contributions of—and potential interplay between—common variants, rare variants, and environmental factors in the pathogenesis of DCM.”

We have also included **Supplemental Table 13** at the end of this document.

Page 14 ll. 282-297 – As noted above, without formal mapping of lead SNPs to likely causal genes the claims in this paragraph seem overstated. While it is possible, or even likely, that the lead SNPs act through *FLNC*, *SSPN* and *PXN* the authors have not sought to confirm this and the wording should be moderated to be more circumspect.

Author response:

We appreciate the Reviewer's helpful suggestion to implement a more formal approach to mapping. We have now conducted a TWAS as an expression-based gene prioritization strategy. Of the three genes discussed in the costamere section of the discussion, TWAS prioritized both *SSPN* and *PXN* at their respective loci. *FLNC* was excluded from the TWAS due to being below the GTEx v7 inclusion threshold for eQTL P values in the two available cardiac tissues (the left ventricle and the right atrial appendage).

Manuscript change:

We have noted the TWAS results in the **Discussion**:

“Notably, the genes closest to four lead SNPs (*FLNC*, *PTK2*, *PXN*, and *SSPN*) play key roles in costamere biology, and two (*SSPN* and *PXN*) are also the most strongly prioritized genes at their respective loci by TWAS.”

We have also moderated our language in the Discussion paragraph about the costamere:

“The emergence of these loci from our unbiased genetic analysis of cardiac MRI phenotypes points to the importance of the costamere within the cardiomyocyte, and suggests that compromise of its protein assemblies may contribute to the development of cardiomyopathies.”

Modified tables & figures

Table 1: Genome-wide significant loci from genetic association analyses of cardiac MRI phenotypes

Loc ID	dbSNP	Ch	Position (hg19)	Effect Allele	Other Allele	EAF	INFO	Beta	SE	P value	Nearest Gene	TWAS Gene	Cardiac Disease Gene	Prior
LVEDV														
1	rs28579893	1	16347534	A	G	0.33	1.00	-0.037	0.0062	1.00E-09	CLCNKA	RP11-169K16.8	PLEKHM2	E, V
2	rs753562515	1	46007032	CAA	C	0.56	0.99	-0.035	0.0059	4.40E-09	AKR1A1	CCDC17		
3	rs7605066	2	71529331	C	T	0.43	1.00	0.033	0.0059	1.60E-08	ZNF638	RP11-467P9.1		
4	rs539762056	2	174892676	A	AT	0.65	0.95	0.035	0.0063	3.40E-08	SP3	RP11-394I13.2		
5	rs1873164	2	179753549	G	A	0.20	0.99	-0.060	0.0073	1.20E-16	CCDC141	RP11-17112.1	TTN	E, A
6	rs73028849	3	14272766	G	C	0.66	0.99	0.040	0.0062	2.10E-10	XPC	SLC6A6	TMEM43	
7	rs6777123	3	169303070	A	C	0.39	0.99	0.034	0.0060	1.80E-08	MECOM	RP11-362K14.5		
8	rs4521636	6	31312293	T	C	0.47	1.00	0.039	0.0059	1.10E-10	HLA-B	NA		M
9	rs9275587	6	32680379	T	C	0.52	0.98	0.037	0.0060	8.90E-10	HLA-DQA2	NA		
10	rs2146324	6	43756863	A	C	0.26	0.98	0.039	0.0068	1.00E-08	VEGFA	RSPH9		
11	rs11153730	6	118667522	T	C	0.51	1.00	-0.048	0.0059	3.70E-16	PLN	SSXP10	PLN	W
12	rs3918226	7	150690176	C	T	0.92	0.97	0.058	0.0110	4.90E-08	NOS3	NUB1		
13		8	125858538	GA	G	0.69	0.98	0.042	0.0064	1.80E-11	MTSS1	MTSS1		W, K
14	rs72840788	10	121415685	G	A	0.79	0.98	0.055	0.0072	1.50E-14	BAG3	GRK5	BAG3	E, V, A
15	rs7306710	12	66376091	T	C	0.48	0.99	0.039	0.0060	5.90E-11	LLPH	LLPH		
16	rs3184504	12	111884608	T	C	0.47	1.00	-0.048	0.0059	3.30E-16	SH2B3	SH2B3		W, A
17	rs10850034	12	112817521	T	A	0.65	0.87	0.050	0.0066	7.60E-15	HECTD4	PTPN11	PTPN11	
18		15	85348961	TTTTGT	T	0.75	0.97	-0.040	0.0068	2.80E-09	ZNF592	ALPK3	ALPK3	E
19	rs71385734	16	2160503	T	G	0.83	0.99	0.050	0.0079	2.10E-10	PKD1	TBC1D24		
20	rs2302455	17	1374195	G	A	0.88	0.97	0.059	0.0093	1.30E-10	MYO1C	TLCD2		
21		19	41945122	CT	C	0.61	0.99	-0.037	0.0060	6.80E-10	ATP5SL	C19orf69		
22	rs12460541	19	46312077	G	A	0.65	1.00	-0.040	0.0062	6.90E-11	RSPH6A	DMPK		
LVEDVi														
1	rs1976402	1	16143779	G	A	0.71	1.00	0.043	0.0077	1.10E-08	SPEN	RP11-169K16.8	PLEKHM2	E, V
5	rs1873164	2	179753549	G	A	0.20	0.99	-0.070	0.0088	1.80E-15	CCDC141	RP11-17112.1	TTN	E, A
23	rs767987273	2	201170509	CA	C	0.60	0.94	0.040	0.0074	4.80E-08	SPATS2L	SPATS2L		
6	rs73028849	3	14272766	G	C	0.66	0.99	0.044	0.0074	4.10E-09	XPC	RP11-536I6.1	TMEM43	
24	rs12499670	4	174621493	T	C	0.40	0.97	0.040	0.0073	3.60E-08	HAND2	RP11-475B2.1		
10	rs6458349	6	43759789	G	A	0.27	1.00	0.044	0.0080	1.50E-08	VEGFA	RSPH9		
25	rs9480737	6	107442277	A	G	0.68	1.00	-0.043	0.0076	2.30E-08	BEND3	BEND3		
11	rs72967533	6	118655020	T	C	0.52	0.99	-0.049	0.0071	7.10E-12	PLN	SSXP10	PLN	W
13		8	125858538	GA	G	0.69	0.98	0.045	0.0077	2.30E-09	MTSS1	MTSS1		W, K
14	rs72840788	10	121415685	G	A	0.79	0.98	0.059	0.0087	1.00E-11	BAG3	GRK5	BAG3	E, V, A
16	rs35350651	12	111907431	A	AC	0.48	1.00	-0.044	0.0071	2.40E-10	ATXN2	RP3-462E2.5		W, A
17	rs10850034	12	112817521	T	A	0.65	0.87	0.042	0.0079	3.90E-08	HECTD4	PTPN11	PTPN11	
20	rs7502466	17	1372970	G	A	0.89	0.98	0.063	0.0114	2.30E-08	MYO1C	SLC43A2		
22	rs9797817	19	46312345	C	T	0.65	1.00	-0.051	0.0074	4.60E-12	RSPH6A	ERCC2		
LVEF														
26	rs2503715	1	2144107	A	G	0.13	0.91	-0.058	0.0108	4.00E-08	C1orf86	SKI		
1	rs1739837	1	16337933	C	T	0.41	1.00	0.071	0.0070	2.90E-25	HSPB7	RP11-169K16.8	PLEKHM2	E, V, A
27	rs10925197	1	236842077	C	G	0.46	0.99	0.042	0.0069	6.90E-10	ACTN2	ACTN2	ACTN2, RYR2	
5	rs2562845	2	179514433	T	C	0.80	1.00	-0.079	0.0086	5.70E-20	TTN	RP11-17112.1	TTN	E, A
6	rs11710541	3	14291679	T	C	0.66	0.99	-0.065	0.0073	9.10E-20	XPC	RP11-536I6.1	TMEM43	
28	rs56099248	3	69857773	C	T	0.81	1.00	-0.061	0.0087	5.10E-12	MITF	NA		
29	rs35999985	5	138756825	A	G	0.30	1.00	0.048	0.0076	3.80E-10	DNAJC18	SPATA24		
9	rs9274626	6	32636040	T	C	0.32	0.95	-0.042	0.0076	2.70E-08	HLA-DQB1	NA		

30	rs3176326	6	36647289	G	A	0.80	0.99	-0.078	0.0087	1.80E-19	CDKN1A	CDKN1A			W
31	rs3807309	7	128472138	G	A	0.89	0.98	-0.083	0.0109	4.10E-15	FLNC	CCDC136	FLNC		E
32	rs36029352	8	11786925	C	T	0.45	0.99	-0.038	0.0070	1.30E-08	DEFB136	FDFT1	GATA4		
33	rs4073554	8	141704232	T	C	0.48	0.98	0.043	0.0070	1.10E-09	PTK2	CTD-3064M3.4			
34	rs189569984	10	112544125	C	T	0.99	1.00	-0.201	0.0375	3.40E-08	RBM20	SHOC2	RBM20, SHOC2		
14	rs72840788	10	121415685	G	A	0.79	0.98	-0.100	0.0085	4.00E-32	BAG3	GRK5	BAG3		E, V, A
35	rs721067	11	19212726	T	A	0.92	0.98	0.070	0.0127	3.60E-08	CSRP3	E2F8	CSRP3		
36	rs113819537	12	26348429	C	G	0.75	0.99	0.045	0.0080	1.70E-08	SSPN	SSPN			
18	rs8023658	15	85323220	G	T	0.51	0.96	-0.046	0.0071	4.60E-11	ZNF592	ALPK3	ALPK3		E
37	rs5029142	16	988070	T	A	0.62	0.99	-0.043	0.0072	3.90E-10	LMF1	METRN			
38	rs12452367	17	53374610	T	C	0.71	0.98	-0.055	0.0077	1.30E-13	HLF	HLF			
39	rs2047273	18	34184859	T	C	0.68	0.96	-0.042	0.0075	3.10E-08	FHOD3	FHOD3			E
40	rs10871753	18	55956865	G	T	0.49	0.99	-0.043	0.0069	6.10E-10	NEDD4L	RP11-845C23.2			
41	rs2070458	22	24159307	A	T	0.20	1.00	0.063	0.0087	2.50E-13	SMARCB1	MMP11			K
LVESV															
42	rs114300540	1	6248182	C	T	0.88	0.90	-0.059	0.0096	3.20E-10	RPL22	ICMT			
1	rs1048302	1	16340879	T	G	0.33	1.00	-0.060	0.0064	4.40E-22	HSPB7	RP11-169K16.8	PLEKHM2		E, V
2	rs753562515	1	46007032	CAA	C	0.56	0.99	-0.037	0.0061	3.40E-09	AKR1A1	MUTYH			
5	rs2562845	2	179514433	T	C	0.80	1.00	0.077	0.0076	1.30E-23	TTN	RP11-17112.1	TTN		E, A
6	rs11710541	3	14291679	T	C	0.66	0.99	0.058	0.0064	4.90E-20	XPC	RP11-53616.1	TMEM43		
43	rs13092177	3	134455794	G	T	0.85	0.99	0.048	0.0085	1.30E-08	EPHB1	EPHB1			
44	rs1499813	3	171760427	T	C	0.59	1.00	0.035	0.0062	6.90E-09	FNDC3B	TNFSF10			
9	rs9274626	6	32636040	T	C	0.32	0.95	0.039	0.0067	2.90E-09	HLA-DQB1	NA			
30	rs730506	6	36645968	G	C	0.80	0.99	0.053	0.0076	2.10E-12	CDKN1A	CDKN1A			
11	rs11153730	6	118667522	T	C	0.51	1.00	-0.038	0.0061	6.80E-10	PLN	SSXP10	PLN		
31	rs34373805	7	128486363	C	T	0.84	1.00	0.055	0.0083	9.20E-12	FLNC	CCDC136	FLNC		E
13		8	125858538	GA	G	0.69	0.98	0.048	0.0066	2.40E-13	MTSS1	MTSS1			K, A
33	rs1962104	8	141635329	T	C	0.45	0.98	-0.037	0.0061	2.20E-09	AGO2	CTD-3064M3.4			
14	rs72840788	10	121415685	G	A	0.79	0.98	0.087	0.0075	8.10E-32	BAG3	GRK5	BAG3		E, V, A
45	rs10832164	11	14048480	C	T	0.49	1.00	-0.033	0.0061	3.80E-08	RRAS2	SPON1			
35	rs11604807	11	19231167	T	C	0.86	0.99	-0.049	0.0088	3.50E-08	CSRP3	E2F8	CSRP3		
36	rs113819537	12	26348429	C	G	0.75	0.99	-0.040	0.0070	1.30E-08	SSPN	SSPN			
16	rs3184504	12	111884608	T	C	0.47	1.00	-0.039	0.0061	1.10E-10	SH2B3	RP3-462E2.5			
17	rs10850034	12	112817521	T	A	0.65	0.87	0.041	0.0068	7.60E-10	HECTD4	PTPN11	PTPN11		
46	rs116904997	12	120668534	G	A	0.98	0.99	-0.112	0.0206	3.40E-08	PXN	PXN			
18		15	85348961	TTTTG	T	0.75	0.97	-0.051	0.0071	1.30E-13	ZNF592	ALPK3	ALPK3		E
37	rs3829491	16	1004834	T	C	0.61	1.00	0.034	0.0062	9.60E-09	LMF1	METRN			
19	rs71385734	16	2160503	T	G	0.83	0.99	0.053	0.0081	4.50E-11	PKD1	TBC1D24			
20	rs2302455	17	1374195	G	A	0.88	0.97	0.063	0.0096	8.60E-11	MYO1C	TLCD2			
47	rs242562	17	44026739	G	A	0.62	0.97	-0.037	0.0063	3.90E-09	MAPT	KANSL1			
38	rs12452367	17	53374610	T	C	0.71	0.98	0.047	0.0068	5.60E-13	HLF	HLF			
48	rs9897002	17	64286494	A	G	0.57	0.99	0.033	0.0061	4.00E-08	PRKCA	PRKCA			
40	rs10871753	18	55956865	G	T	0.49	0.99	0.033	0.0061	3.00E-08	NEDD4L	RP11-845C23.2			
49		19	10765478	CG	C	0.24	0.98	0.041	0.0072	1.20E-08	ILF3	SMARCA4			
21	rs16975238	19	41944985	T	A	0.60	1.00	-0.034	0.0062	2.00E-08	ATP5SL	EXOSC5			
22	rs10421891	19	46315809	A	G	0.65	1.00	-0.045	0.0063	6.70E-13	RSPH6A	AC074212.3			
41	rs5760061	22	24178279	G	A	0.20	1.00	-0.055	0.0076	1.30E-13	DERL3	MMP11			K
LVESVi															
42	rs709208	1	6272137	A	G	0.68	0.95	-0.041	0.0075	1.90E-08	RNF207	ICMT			
1	rs945425	1	16348412	T	C	0.32	1.00	-0.070	0.0073	8.50E-23	CLCNKA	RP11-169K16.8	PLEKHM2		E, V
2	rs753562515	1	46007032	CAA	C	0.56	0.99	-0.038	0.0069	1.70E-08	AKR1A1	MUTYH			
5	rs2562845	2	179514433	T	C	0.80	1.00	0.085	0.0086	1.60E-22	TTN	RP11-17112.1	TTN		E, A
50	rs190093681	2	180094352	C	T	1.00	0.84	0.374	0.0651	1.60E-08	SESTD1	RP11-17112.1			
23	rs774290282	2	201198623	CATT	C	0.61	1.00	-0.038	0.0070	4.70E-08	SPATS2L	SPATS2L			
6	rs73028849	3	14272766	G	C	0.66	0.99	0.064	0.0073	6.20E-19	XPC	RP11-53616.1	TMEM43		
28	rs79502300	3	69856753	C	T	0.80	1.00	0.054	0.0087	4.40E-10	MITF	NA			

51	rs2886037	3	158306414	G	A	0.49	0.96	-0.036	0.0070	2.90E-08	MLF1	MLF1		
44	rs1499813	3	171760427	T	C	0.59	1.00	0.040	0.0070	2.80E-09	FNDC3B	AC092964.2		
29	rs11748963	5	138730037	T	C	0.73	0.99	0.044	0.0077	5.90E-09	PROB1	PROB1		
30	rs3176326	6	36647289	G	A	0.80	0.99	0.065	0.0087	1.80E-14	CDKN1A	CDKN1A		
31	rs34373805	7	128486363	C	T	0.84	1.00	0.067	0.0094	2.70E-13	FLNC	CCDC136	FLNC	E
13		8	125858538	GA	G	0.69	0.98	0.051	0.0075	3.30E-12	MTSS1	MTSS1		K, A
33	rs1962104	8	141635329	T	C	0.45	0.98	-0.043	0.0069	6.40E-10	AGO2	CTD-3064M3.7		
34	rs189569984	10	112544125	C	T	0.99	1.00	0.214	0.0369	2.10E-09	RBM20	SHOC2	RBM20, SHOC2	
14	rs72840788	10	121415685	G	A	0.79	0.98	0.093	0.0084	1.70E-28	BAG3	GRK5	BAG3	E, V, A
45	rs11023059	11	14064392	A	G	0.52	1.00	0.042	0.0069	6.50E-10	RRAS2	SPON1		
46	rs116904997	12	120668534	G	A	0.98	0.99	-0.132	0.0233	1.10E-08	PXN	PXN		
18		15	85348961	TTTTGT	T	0.75	0.97	-0.054	0.0080	1.40E-11	ZNF592	NMB	ALPK3	E
37	rs8063213	16	992961	T	G	0.62	1.00	0.045	0.0071	7.60E-11	LMF1	LMF1		
20	rs2302455	17	1374195	G	A	0.88	0.97	0.065	0.0109	4.60E-09	MYO1C	TLCD2		
47	rs242562	17	44026739	G	A	0.62	0.97	-0.040	0.0072	3.00E-08	MAPT	KANSL1		
38	rs12452367	17	53374610	T	C	0.71	0.98	0.055	0.0077	1.80E-13	HLF	HLF		
48	rs9892651	17	64303793	C	T	0.42	1.00	0.040	0.0070	2.10E-09	PRKCA	PRKCA		
40	rs10871753	18	55956865	G	T	0.49	0.99	0.039	0.0069	6.00E-09	NEDD4L	RP11-845C23.2		
22	rs10421891	19	46315809	A	G	0.65	1.00	-0.053	0.0072	1.20E-13	RSPH6A	AC074212.3		
41	rs5760061	22	24178279	G	A	0.20	1.00	-0.064	0.0086	7.50E-15	DERL3	MMP11		K
SV														
5	rs7573293	2	179753245	C	T	0.27	0.99	-0.049	0.0071	2.00E-12	CCDC141	RP11-17112.1	TTN	E
52	rs888690	5	172636130	T	C	0.40	0.98	-0.039	0.0065	1.20E-09	NKX2-5	NKX2-5	NKX2-5	
8	rs111721712	6	31315407	C	CT	0.53	0.99	0.038	0.0063	3.40E-09	HLA-B	NA		M
9	rs28391274	6	32623786	A	G	0.80	0.85	0.047	0.0084	1.40E-08	HLA-DQB1	NA		
10	rs2146324	6	43756863	A	C	0.26	0.98	0.041	0.0072	1.10E-08	VEGFA	RSPH9		
11	rs72967533	6	118655020	T	C	0.52	0.99	-0.051	0.0063	1.40E-15	PLN	SSXP10	PLN	
15	rs10400419	12	66389968	T	C	0.45	0.95	0.040	0.0065	5.40E-10	LLPH	LLPH		
16	rs11065979	12	112059557	C	T	0.57	1.00	0.048	0.0064	3.90E-14	ATXN2	ALDH2		
17	rs11066188	12	112610714	G	A	0.60	1.00	0.045	0.0064	2.70E-12	HECTD4	PTPN11	PTPN11	
17	rs2891403	12	113137572	A	G	0.28	0.98	-0.042	0.0071	3.80E-09	RPH3A	PTPN11	PTPN11	
53	rs422068	14	23864804	T	C	0.64	1.00	0.039	0.0065	1.40E-09	MYH6	CEBPE	MYH6, MYH7	
54	rs143384	20	34025756	A	G	0.59	1.00	-0.035	0.0064	3.60E-08	GDF5	UQC1		
SVI														
5	rs7573293	2	179753245	C	T	0.28	0.99	-0.051	0.0082	7.10E-10	CCDC141	RP11-17112.1	TTN	E
55		2	232288831	TTTC	T	0.68	0.99	-0.045	0.0079	9.30E-09	B3GNT7	B3GNT7		
10	rs6458349	6	43759789	G	A	0.27	1.00	0.045	0.0082	3.50E-08	VEGFA	RSPH9		
25	rs9480737	6	107442277	A	G	0.68	1.00	-0.047	0.0078	2.60E-09	BEND3	C6orf203		
11	rs72967533	6	118655020	T	C	0.52	0.99	-0.048	0.0073	1.10E-10	PLN	PLN	PLN	
56	rs1919865	6	122121005	A	T	0.86	0.98	0.060	0.0108	1.90E-08	GJA1	GJA1		
57	rs579459	9	136154168	T	C	0.79	1.00	-0.050	0.0090	4.10E-08	SURF6	SURF2		
53	rs376439	14	23869029	A	G	0.60	0.99	0.043	0.0075	6.50E-09	MYH6	MYH6	MYH6, MYH7	

Loc ID: Each distinct genomic locus is labeled with a number from 1-40 for tracking across phenotypes. *dbSNP*: For variants which were assigned an rsID by dbSNP as of version 151, that rsID is listed. *Ch*: Chromosome. *EAF*: Effect allele frequency. *INFO*: Information score provided by the UK Biobank describing the quality of imputation at each locus from IMPUTE2. This is set to a value of 1 for directly genotyped SNPs. *Nearest Gene*: Gene closest to the lead SNP. *TWAS Gene*: Gene within 500kb of the lead SNP with the lowest TWAS P value at the locus (if any). *Cardiac Disease Genes*: Genes within 500kb of the lead SNP that have previously been linked to DCM or another Mendelian cardiomyopathy from the combined gene panel in **Supplemental Table 7**. *Prior*: Symbols represent prior studies that had linked the locus to an analogous echocardiographic trait (K = Kanai, et al; W = Wild, et al; A = Aung, et al) or dilated cardiomyopathy (V = Villard, et al; E = Esslinger, et al; M = Meder, et al)(Aung Nay et al., 2019; Esslinger et al., 2017, 2017; Kanai et al., 2018; Meder et al., 2014; Wild et al., 2017). In total, 57 loci are identified, of which 12 are annotated as having previously been associated with

cardiac structure and function or dilated cardiomyopathy. The effect size and standard error are dimensionless due to the inverse normal transform; a value of 1 represents 1 standard deviation from the mean.

Figure 2: PheWAS highlights the connection between a polygenic score for LVESVi and dilated cardiomyopathy

The polygenic score derived from LVESVi was applied to PheCodes (panel a) and curated disease phenotypes (panel b) in the UK Biobank. Each of the curated phenotypes is defined in **Supplemental Table 1**. For both panels (a) and (b), the X-axis represents the identifying code

for the disease phenotype. The Y-axis represents the $-\log_{10}$ of the P value of association between the polygenic score and the phenotype in a logistic model adjusted for age at enrollment, the genotyping array, sex, and the first five principal components of ancestry. Triangles oriented upward represent betas that are concordant with the LVESVi PRS (e.g., a higher LVESVi PRS corresponds with a higher risk of DCM), and the reverse is true for downward-oriented triangles. The 3 most strongly associated phenotypes in each panel are labeled on the figure. The triangles in panel (a) are colored by 3-digit PheCode. The triangles in panel (b) are colored red if positively correlated with the LVESVi polygenic score, and blue if negatively correlated. The PheWAS plots for all seven cardiac MRI phenotypes are available in **Supplemental Figure 7**.

Supplemental Table 4: Lookup of SNPs from prior GWAS studies in the cardiac MRI GWAS

Study	Discovery Trait	SNP	CHR	BP_HG19	Nearby Genes	Locus	Discovery P	MRI Trait	MRI GWAS P
Kanai	LVIDd	rs34866937	8	125859850	MTSS1	TTE 1	1.77E-09	LVEDV	6.20E-11
Kanai	LVIDd	rs3812625	10	75757702	VCL	TTE 2	1.42E-08	LVEDV	6.00E-01
Kanai	LVIDd	rs11874741	18	30077859	GAREM1	TTE 3	4.94E-08	LVEDV	9.70E-01
Kanai	LVIDs	rs6546120	2	65238407	SLC1A4	TTE 4	4.79E-08	LVESV	4.90E-01
Kanai	LVIDs	rs34866937	8	125859850	MTSS1	TTE 1	1.60E-13	LVESV	8.70E-13
Kanai	LVIDs	rs11874741	18	30077859	GAREM1	TTE 3	4.62E-08	LVESV	9.90E-01
Kanai	LVIDs	rs5760061	22	24178279	SMARCB1/DERL3	TTE 5	8.63E-11	LVESV	1.30E-13
Kanai	Fractional Shortening	rs6546120	2	65238407	SLC1A4	TTE 4	1.70E-08	LVEF	7.20E-01
Kanai	Fractional Shortening	rs34866937	8	125859850	MTSS1	TTE 1	2.40E-13	LVEF	1.40E-07
Kanai	Fractional Shortening	rs11025521	11	20370206	DBX1	TTE 6	4.41E-08	LVEF	6.80E-02
Kanai	Fractional Shortening	rs5760054	22	24161717	SMARCB1/DERL3	TTE 5	1.01E-11	LVEF	2.70E-13
Kanai	Ejection Fraction	rs6546120	2	65238407	SLC1A4	TTE 4	3.35E-08	LVEF	7.20E-01
Kanai	Ejection Fraction	rs34866937	8	125859850	MTSS1	TTE 1	8.97E-16	LVEF	1.40E-07
Kanai	Ejection Fraction	rs5760061	22	24178279	SMARCB1/DERL3	TTE 5	6.84E-11	LVEF	2.80E-13
Wild	LVIDd	rs11153730	6	118667522	SLC35F1/PLN	TTE 7	6.40E-16	LVEDV	3.70E-16
Wild	LVIDd	rs12541595	8	125857359	MTSS1	TTE 1	3.02E-12	LVEDV	1.80E-10
Wild	LVIDd	rs10774625	12	111910219	ATXN2	TTE 8	1.90E-08	LVEDV	1.70E-15
Wild	Fractional Shortening	rs9470361	6	36623379	CDKN1A	TTE 9	5.30E-09	LVEF	2.00E-15
Wild	LV Mass	rs1454157	4	177358798	SPCS3	TTE 10	4.41E-09		
Esslinger	DCM	rs10927875	1	16299312	CLCNKA	DCM 1	8.11E-13	LVEF	3.10E-21
Esslinger	DCM	rs848210	1	16259813	CLCNKA	DCM 1	6.30E-07	LVEF	4.30E-20
Esslinger	DCM	rs3829746	2	179427536	TTN	DCM 2	3.40E-07	LVESV	2.30E-19
Esslinger	DCM	rs13107325	4	103188709	SLC39A8	DCM 3	6.00E-07	LVESV	1.10E-04
Esslinger	DCM	rs4712056	6	53989526	MLIP	DCM 4	5.10E-07	LVEDV	1.50E-06
Esslinger	DCM	rs2291569	7	128488734	FLNC	DCM 5	8.70E-11	LVESV	2.80E-10
Esslinger	DCM	rs2234962	10	121429633	BAG3	DCM 6	1.70E-25	LVESV	1.80E-30
Esslinger	DCM	rs3188055	10	121586882	BAG3	DCM 6	1.10E-08	LVEF	8.60E-06
Esslinger	DCM	rs1051168	15	85200520	ALPK3	DCM 7	4.10E-07	LVESVi	6.90E-10
Esslinger	DCM	rs3803403	15	85383145	ALPK3	DCM 7	2.90E-07	LVESV	1.20E-11
Esslinger	DCM	rs2303510	18	34324091	FHOD3	DCM 8	1.50E-07	LVEF	6.10E-06 ^a
Meder	DCM	rs9262636	6	31025848	HCG22	DCM 9	4.90E-09	LVEDVi	4.30E-01
Villard	DCM	rs10927875	1	16299312	CLCNKA	DCM 1	1.30E-07	LVEF	3.10E-21
Villard	DCM	rs2234962	10	121429633	BAG3	DCM 6	8.80E-10	LVEF	5.40E-31
Aung	LVEDV	rs2042995	2	179558366	TTN	AUN 1	2.30E-11	LVEDV	1.50E-14
Aung	LVEDV	rs7071853	10	121311606	BAG3	AUN 2	3.90E-09	LVEDV	1.70E-05
Aung	LVEDV	rs7310615	12	111865049	SH2B3	AUN 3	1.40E-09	LVEDV	4.90E-16
Aung	LVESV	rs2042995	2	179558366	TTN	AUN 4	8.40E-20	LVESV	5.90E-23
Aung	LVESV	rs200712209	8	125858538	MTSS1	AUN 5	1.70E-11	LVESV	2.70E-13
Aung	LVESV	rs72840788	10	121415685	BAG3	AUN 6	5.60E-17	LVESV	8.10E-32
Aung	LVEF	rs945425	1	16348412	CLCNKA	AUN 7	8.60E-11	LVEF	6.00E-25
Aung	LVEF	rs2042995	2	179558366	TTN	AUN 8	2.50E-12	LVEF	1.30E-18
Aung	LVEF	rs34866937	8	125859850	MTSS1	AUN 9	6.80E-11	LVEF	1.40E-07

Aung	LVEF	rs72840788	10	121415685	BAG3	AUN 10	3.40E-15	LVEF	4.00E-32
Aung	LV Mass	rs2255167	2	179558282	TTN	AUN 11	8.30E-14		
Aung	LV Mass-Volume Ratio	rs146170154	6	36646768	CDKN1A	AUN 12	2.60E-11		
Aung	LV Mass-Volume Ratio	rs149369954	15	85348961	ZNF592/ALPK3	AUN 13	1.90E-11		
Aung	LV Mass-Volume Ratio	rs6003909	22	24181652	DERL3	AUN 14	9.70E-15		

Study: The discovery study in which the SNPs were identified: Kanai, Wild, Esslinger, Meder, Villard, Aung¹⁻⁶. LVIDd: left ventricular internal diameter at end diastole from transthoracic echocardiography (TTE). LVIDs: left ventricular diameter at end systole from TTE. LV Mass: left ventricular mass from TTE. DCM: dilated cardiomyopathy. The “Locus” column tracks unique genomic loci, such that, within a study type (TTE or DCM), SNPs within 1 megabase of one another are given the same Locus identifier. For SNPs from the studies of cardiac traits, the most similar MRI trait was used for comparison. No similar trait was available for comparison to LV Mass. For SNPs from the DCM GWAS studies, the MRI trait with the lowest P-value was taken for each SNP. The “MRI GWAS P” column represents the association P value of the SNP from the SNP column within in the cardiac MRI GWAS from the “MRI Trait” column. The MRI GWAS P value column is shaded blue if the SNP had $P < 5 \times 10^{-8}$ in the MRI GWAS, and yellow if the SNP had $5 \times 10^{-8} < P < 1 \times 10^{-3}$. The values represent the P value for the specific SNP, but other SNPs in linkage disequilibrium may have stronger association P values. Note: (a) At rs2303510 (DCM 8), the nearby SNP rs2047273 ($r^2 = 0.29$) in the locus had an association $P = 3.1 \times 10^{-8}$ with LVEDV. After accounting for SNPs in linkage disequilibrium, five of the 10 TTE loci and six of the nine DCM loci have a genome-wide significant SNP in the cardiac MRI GWAS.

Supplemental Table 5: Characteristics of MESA Participants

		Women	Men	All
		1204	980	2184
Age at MRI				
	Mean (SD)	69 (9.1)	68 (8.9)	69 (9.1)
	Median [Q1-Q3]	68 [61 - 76]	67 [61 - 75]	68 [61 - 75]
	Missing	5 (0.4%)	3 (0.3%)	8 (0.4%)
Site				
	Wake Forest	197 (16 %)	135 (14 %)	332 (15 %)
	Columbia	200 (17 %)	165 (17 %)	365 (17 %)
	Johns Hopkins	181 (15 %)	127 (13 %)	308 (14 %)
	Minnesota	173 (14 %)	160 (16 %)	333 (15 %)
	Northwestern	243 (20 %)	186 (19 %)	429 (20 %)
	UCLA	210 (17 %)	207 (21 %)	417 (19 %)
Ancestry				
	European	537 (45 %)	403 (41 %)	940 (43 %)
	East Asian	156 (13 %)	143 (15 %)	299 (14 %)
	African-American	297 (25 %)	210 (21 %)	507 (23 %)
	Hispanic	214 (18 %)	224 (23 %)	438 (20 %)
BMI (kg/cm²)				
	Mean (SD)	28 (5.4)	27 (4.0)	28 (4.8)
	Median [Q1-Q3]	27 [24 - 31]	27 [25 - 30]	27 [24 - 30]
Weight (kg)				
	Mean (SD)	72 (15)	83 (14)	77 (16)
	Median [Q1-Q3]	69 [61 - 81]	83 [73 - 92]	75 [65 - 87]
Height (cm)				

	Mean (SD)	160 (6.8)	170 (7.6)	170 (9.7)
	Median [Q1-Q3]	160 [160 - 170]	170 [170 - 180]	170 [160 - 170]
LVEDV				
	Mean (SD)	110 (23)	130 (32)	120 (31)
	Median [Q1-Q3]	100 [90 - 120]	130 [110 - 150]	110 [96 - 140]
LVEDVi				
	Mean (SD)	59 (11)	67 (14)	63 (13)
	Median [Q1-Q3]	59 [52 - 66]	66 [57 - 76]	62 [54 - 70]
LVESV				
	Mean (SD)	38 (12)	54 (18)	45 (17)
	Median [Q1-Q3]	36 [30 - 45]	51 [41 - 63]	42 [33 - 54]
LVESVi				
	Mean (SD)	21 (6.0)	27 (8.1)	24 (7.5)
	Median [Q1-Q3]	21 [17 - 25]	26 [21 - 31]	23 [19 - 28]
LVEF				
	Mean (SD)	64 (6.6)	60 (6.7)	62 (7.0)
	Median [Q1-Q3]	65 [60 - 69]	60 [56 - 65]	63 [58 - 67]
SV				
	Mean (SD)	67 (15)	80 (19)	73 (18)
	Median [Q1-Q3]	67 [57 - 76]	78 [66 - 93]	71 [61 - 84]
SVi				
	Mean (SD)	38 (7.5)	40 (9.1)	39 (8.3)
	Median [Q1-Q3]	38 [33 - 43]	40 [34 - 46]	39 [33 - 44]

MESA participants with both whole genome sequencing data and cardiac MRI at “Exam 5” were included in the analysis, excluding those with heart disease or evidence of myocardial scar.

Supplemental Table 6: External validation

Trait	SNP	CHR	BP HG19	Effect Allele	Other Allele	EAF	Beta	P value	MESA Sign	MESA P value	BioBank Japan Sign	BioBank Japan P value
lvedv	rs28579893	1	16347534	A	G	0.327	-0.0369	1.00E-09	+	6.63E-01	+	4.49E-01
lvedv	rs753562515	1	46007032	CAA	C	0.561	-0.0353	4.40E-09				
lvedv	rs7605066	2	71529331	C	T	0.429	0.0332	1.60E-08	+	7.27E-02	+	7.09E-01
lvedv	rs539762056	2	174892676	A	AT	0.649	0.0352	3.40E-08				
lvedv	rs1873164	2	179753549	G	A	0.200	-0.0604	1.20E-16	-	9.13E-01	-	8.87E-02
lvedv	rs73028849	3	14272766	G	C	0.658	0.0396	2.10E-10	+	5.35E-01	+	3.01E-04
lvedv	rs6777123	3	169303070	A	C	0.393	0.0335	1.80E-08	+	1.15E-01	+	5.69E-01
lvedv	rs4521636	6	31312293	T	C	0.466	0.0390	1.10E-10	+	2.36E-01		
lvedv	rs9275587	6	32680379	T	C	0.525	0.0370	8.90E-10	+	3.95E-01	-	4.50E-01
lvedv	rs2146324	6	43756863	A	C	0.261	0.0386	1.00E-08	+	6.53E-01	-	5.25E-01
lvedv	rs11153730	6	118667522	T	C	0.513	-0.0482	3.70E-16	+	1.70E-01	+	1.94E-03
lvedv	rs3918226	7	150690176	C	T	0.921	0.0585	4.90E-08	+	1.06E-01		
lvedv		8	125858538	GA	G	0.688	0.0422	1.80E-11	-	7.43E-01		
lvedv	rs72840788	10	121415685	G	A	0.785	0.0549	1.50E-14	+	9.58E-03		
lvedv	rs7306710	12	66376091	T	C	0.480	0.0391	5.90E-11	+	2.17E-01	+	8.50E-01
lvedv	rs3184504	12	111884608	T	C	0.470	-0.0477	3.30E-16	+	3.50E-01		
lvedv	rs10850034	12	112817521	T	A	0.647	0.0504	7.60E-15				
lvedv		15	85348961	TTTTG	T	0.746	-0.0400	2.80E-09				
lvedv	rs71385734	16	2160503	T	G	0.828	0.0504	2.10E-10	+	1.08E-01		
lvedv	rs2302455	17	1374195	G	A	0.884	0.0591	1.30E-10	+	4.47E-03	-	8.11E-01
lvedv		19	41945122	CT	C	0.606	-0.0371	6.80E-10				
lvedv	rs12460541	19	46312077	G	A	0.653	-0.0398	6.90E-11	+	2.24E-02	+	1.88E-01
lvedvi	rs1976402	1	16143779	G	A	0.712	0.0435	1.10E-08	+	1.46E-01		
lvedvi	rs1873164	2	179753549	G	A	0.200	-0.0701	1.80E-15	+	9.30E-01		

lvedvirs767987273	2	201170509CA	C	0.602	0.0399	4.80E-08				
lvedvirs73028849	3	14272766G	C	0.659	0.0439	4.10E-09	+	7.14E-01		
lvedvirs12499670	4	174621493T	C	0.396	0.0401	3.60E-08	+	5.90E-01		
lvedvirs6458349	6	43759789G	A	0.272	0.0444	1.50E-08	+	1.56E-01		
lvedvirs9480737	6	107442277A	G	0.683	-0.0427	2.30E-08	+	3.47E-01		
lvedvirs72967533	6	118655020T	C	0.521	-0.0488	7.10E-12	+	1.57E-01		
lvedvi	8	125858538GA	G	0.688	0.0455	2.30E-09				
lvedvirs72840788	10	121415685G	A	0.785	0.0588	1.00E-11	+	1.92E-02		
lvedvirs35350651	12	111907431A	AC	0.483	-0.0441	2.40E-10	+	1.99E-01		
lvedvirs10850034	12	112817521T	A	0.648	0.0424	3.90E-08				
lvedvirs7502466	17	1372970G	A	0.891	0.0634	2.30E-08	+	5.99E-02		
lvedvirs9797817	19	46312345C	T	0.653	-0.0508	4.60E-12	+	2.73E-02		
lvef rs2503715	1	2144107A	G	0.128	-0.0584	4.00E-08	+	2.21E-01		
lvef rs1739837	1	16337933C	T	0.411	0.0708	2.90E-25	+	1.26E-01	+	4.16E-02
lvef rs10925197	1	236842077C	G	0.462	0.0420	6.90E-10	+	4.01E-02	-	8.68E-01
lvef rs2562845	2	179514433T	C	0.803	-0.0792	5.70E-20	+	2.50E-01	+	4.04E-05
lvef rs11710541	3	14291679T	C	0.658	-0.0651	9.10E-20	+	3.40E-01	+	2.13E-02
lvef rs56099248	3	69857773C	T	0.805	-0.0607	5.10E-12	+	1.52E-01	+	1.41E-01
lvef rs35999985	5	138756825A	G	0.299	0.0480	3.80E-10	+	3.74E-02	+	2.33E-01
lvef rs9274626	6	32636040T	C	0.319	-0.0417	2.70E-08				
lvef rs3176326	6	36647289G	A	0.800	-0.0777	1.80E-19	+	6.69E-02	+	1.81E-03
lvef rs3807309	7	128472138G	A	0.886	-0.0835	4.10E-15	+	8.25E-01	+	2.05E-01
lvef rs36029352	8	11786925C	T	0.455	-0.0380	1.30E-08				
lvef rs4073554	8	141704232T	C	0.481	0.0428	1.10E-09	-	8.81E-01	+	3.98E-02
lvef rs189569984	10	112544125C	T	0.992	-0.2009	3.40E-08	+	3.13E-03		
lvef rs72840788	10	121415685G	A	0.785	-0.0997	4.00E-32	+	1.45E-02		
lvef rs721067	11	19212726T	A	0.918	0.0697	3.60E-08	-	6.76E-01	+	8.98E-01
lvef rs113819537	12	26348429C	G	0.749	0.0449	1.70E-08	+	1.98E-01	+	2.36E-02
lvef rs8023658	15	85323220G	T	0.507	-0.0464	4.60E-11				
lvef rs5029142	16	988070T	A	0.623	-0.0430	3.90E-10	+	2.51E-01	+	1.76E-01
lvef rs12452367	17	53374610T	C	0.715	-0.0550	1.30E-13	+	7.92E-01	+	7.24E-04
lvef rs2047273	18	34184859T	C	0.678	-0.0418	3.10E-08	+	2.67E-03	-	6.50E-01
lvef rs10871753	18	55956865G	T	0.489	-0.0431	6.10E-10	+	2.67E-01	+	4.57E-03
lvef rs2070458	22	24159307A	T	0.200	0.0629	2.50E-13	+	3.23E-02	+	8.54E-11
lvesv rs114300540	1	6248182C	T	0.875	-0.0590	3.20E-10	-	1.46E-02		
lvesv rs1048302	1	16340879T	G	0.327	-0.0605	4.40E-22	+	2.11E-02	+	4.39E-01
lvesv rs753562515	1	46007032CAA	C	0.561	-0.0365	3.40E-09				
lvesv rs2562845	2	179514433T	C	0.803	0.0770	1.30E-23	+	5.70E-02	+	1.33E-01
lvesv rs11710541	3	14291679T	C	0.658	0.0583	4.90E-20	+	4.12E-01	+	1.13E-04
lvesv rs13092177	3	134455794G	T	0.850	0.0481	1.30E-08	+	4.41E-01	+	2.59E-01
lvesv rs1499813	3	171760427T	C	0.591	0.0351	6.90E-09	+	7.99E-01		
lvesv rs9274626	6	32636040T	C	0.319	0.0393	2.90E-09				
lvesv rs730506	6	36645968G	C	0.800	0.0532	2.10E-12	+	4.19E-02	+	2.53E-01
lvesv rs11153730	6	118667522T	C	0.513	-0.0375	6.80E-10	+	5.21E-01	+	1.94E-03
lvesv rs34373805	7	128486363C	T	0.838	0.0554	9.20E-12	-	5.62E-01	+	2.73E-01
lvesv	8	125858538GA	G	0.688	0.0483	2.40E-13	+	4.67E-01		
lvesv rs1962104	8	141635329T	C	0.449	-0.0371	2.20E-09	+	8.81E-01	+	3.30E-01
lvesv rs72840788	10	121415685G	A	0.785	0.0866	8.10E-32	+	1.49E-03		
lvesv rs10832164	11	14048480C	T	0.486	-0.0333	3.80E-08	+	2.38E-02	+	3.67E-02
lvesv rs11604807	11	19231167T	C	0.862	-0.0493	3.50E-08	-	5.22E-01	+	2.13E-01
lvesv rs113819537	12	26348429C	G	0.749	-0.0395	1.30E-08	+	9.76E-02	+	2.73E-02
lvesv rs3184504	12	111884608T	C	0.470	-0.0388	1.10E-10	+	9.55E-02		
lvesv rs10850034	12	112817521T	A	0.647	0.0411	7.60E-10				
lvesv rs116904997	12	120668534G	A	0.978	-0.1125	3.40E-08	+	5.02E-01		
lvesv	15	85348961TTTTG	T	0.746	-0.0515	1.30E-13				
lvesv rs3829491	16	1004834T	C	0.613	0.0344	9.60E-09			+	4.45E-02
lvesv rs71385734	16	2160503T	G	0.828	0.0531	4.50E-11	+	4.05E-02		

lvesv	rs2302455	17	1374195G	A	0.884	0.0635	8.60E-11	+	4.17E-02	-	8.11E-01
lvesv	rs242562	17	44026739G	A	0.619	-0.0373	3.90E-09	+	6.83E-01		
lvesv	rs12452367	17	53374610T	C	0.715	0.0473	5.60E-13	+	5.16E-01	+	7.76E-03
lvesv	rs9897002	17	64286494A	G	0.571	0.0327	4.00E-08	+	1.78E-02	+	2.53E-01
lvesv	rs10871753	18	55956865G	T	0.489	0.0330	3.00E-08	+	7.25E-01	+	1.44E-01
lvesv		19	10765478CG	C	0.236	0.0408	1.20E-08	+	2.68E-01		
lvesv	rs16975238	19	41944985T	A	0.602	-0.0338	2.00E-08	+	3.59E-01	+	4.44E-02
lvesv	rs10421891	19	46315809A	G	0.645	-0.0451	6.70E-13	+	4.13E-02	+	3.28E-01
lvesv	rs5760061	22	24178279G	A	0.201	-0.0550	1.30E-13	+	1.11E-01	+	2.88E-06
lvesvi	rs709208	1	6272137A	G	0.679	-0.0412	1.90E-08	+	4.63E-01		
lvesvi	rs945425	1	16348412T	C	0.324	-0.0701	8.50E-23	+	1.25E-02		
lvesvi	rs753562515	1	46007032CAA	C	0.561	-0.0384	1.70E-08				
lvesvi	rs2562845	2	179514433T	C	0.804	0.0849	1.60E-22	+	3.03E-02		
lvesvi	rs190093681	2	180094352C	T	0.997	0.3742	1.60E-08	-	2.26E-01		
lvesvi	rs774290282	2	201198623CATT	C	0.608	-0.0377	4.70E-08	-	6.68E-01		
lvesvi	rs73028849	3	14272766G	C	0.659	0.0637	6.20E-19	+	3.52E-01		
lvesvi	rs79502300	3	69856753C	T	0.805	0.0537	4.40E-10	+	5.71E-01		
lvesvi	rs2886037	3	158306414G	A	0.494	-0.0363	2.90E-08				
lvesvi	rs1499813	3	171760427T	C	0.591	0.0398	2.80E-09	+	5.16E-01		
lvesvi	rs11748963	5	138730037T	C	0.726	0.0439	5.90E-09	+	7.86E-02		
lvesvi	rs3176326	6	36647289G	A	0.800	0.0650	1.80E-14	+	1.62E-02		
lvesvi	rs34373805	7	128486363C	T	0.838	0.0672	2.70E-13	-	7.41E-01		
lvesvi		8	125858538GA	G	0.688	0.0507	3.30E-12	+	2.16E-01		
lvesvi	rs1962104	8	141635329T	C	0.449	-0.0427	6.40E-10	+	9.44E-01		
lvesvi	rs189569984	10	112544125C	T	0.991	0.2142	2.10E-09	+	6.09E-02		
lvesvi	rs72840788	10	121415685G	A	0.785	0.0926	1.70E-28	+	1.90E-03		
lvesvi	rs11023059	11	14064392A	G	0.519	0.0416	6.50E-10	+	1.38E-01		
lvesvi	rs116904997	12	120668534G	A	0.978	-0.1316	1.10E-08	+	1.06E-01		
lvesvi		15	85348961TTTTG	T	0.746	-0.0543	1.40E-11				
lvesvi	rs8063213	16	992961T	G	0.618	0.0449	7.60E-11	+	1.19E-01		
lvesvi	rs2302455	17	1374195G	A	0.884	0.0646	4.60E-09	+	2.08E-01		
lvesvi	rs242562	17	44026739G	A	0.619	-0.0397	3.00E-08	+	5.55E-01		
lvesvi	rs12452367	17	53374610T	C	0.715	0.0548	1.80E-13	+	4.69E-01		
lvesvi	rs9892651	17	64303793C	T	0.418	0.0398	2.10E-09	+	1.27E-02		
lvesvi	rs10871753	18	55956865G	T	0.489	0.0394	6.00E-09	+	2.79E-01		
lvesvi	rs10421891	19	46315809A	G	0.646	-0.0528	1.20E-13	+	4.80E-02		
lvesvi	rs5760061	22	24178279G	A	0.201	-0.0645	7.50E-15	+	1.43E-01		
sv	rs7573293	2	179753245C	T	0.275	-0.0492	2.00E-12	-	9.94E-01		
sv	rs888690	5	172636130T	C	0.397	-0.0394	1.20E-09	+	1.83E-01		
sv	rs111721712	6	31315407C	CT	0.527	0.0376	3.40E-09	+	4.52E-01		
sv	rs28391274	6	32623786A	G	0.797	0.0474	1.40E-08				
sv	rs2146324	6	43756863A	C	0.261	0.0411	1.10E-08	+	5.00E-01		
sv	rs72967533	6	118655020T	C	0.521	-0.0511	1.40E-15	+	8.62E-02		
sv	rs10400419	12	66389968T	C	0.448	0.0402	5.40E-10	+	7.39E-02		
sv	rs11065979	12	112059557C	T	0.572	0.0483	3.90E-14	-	7.40E-01		
sv	rs11066188	12	112610714G	A	0.602	0.0450	2.70E-12	-	8.67E-01		
sv	rs2891403	12	113137572A	G	0.278	-0.0418	3.80E-09	+	6.90E-01		
sv	rs422068	14	23864804T	C	0.642	0.0395	1.40E-09	+	1.02E-01		
sv	rs143384	20	34025756A	G	0.588	-0.0349	3.60E-08	+	2.28E-03		
svi	rs7573293	2	179753245C	T	0.275	-0.0506	7.10E-10				
svi		2	232288831TTTC	T	0.684	-0.0455	9.30E-09	+	7.01E-01		
svi	rs6458349	6	43759789G	A	0.272	0.0451	3.50E-08	+	2.02E-01		
svi	rs9480737	6	107442277A	G	0.683	-0.0474	2.60E-09				
svi	rs72967533	6	118655020T	C	0.521	-0.0485	1.10E-10				
svi	rs1919865	6	122121005A	T	0.864	0.0599	1.90E-08				
svi	rs579459	9	136154168T	C	0.794	-0.0500	4.10E-08	-	8.51E-01		
svi	rs376439	14	23869029A	G	0.605	0.0429	6.50E-09				

Trait: Trait tested for replication. CHR: Chromosome. BP HG19: position in Hg19 coordinates. EAF: Effect allele frequency in the UK Biobank primary GWAS. MESA: Multi-Ethnic Study of Atherosclerosis. In the “Sign” columns, a “+” (with cell shaded blue) represents that the effect direction in the UK Biobank primary GWAS for the trait being tested at the SNP was in the same direction as that of the effect in the other cohort; “-” (with cell shaded red) represents that the effects were in opposite directions. Cells without entries (unshaded) represent cells where a matching SNP could not be identified within the external cohort for the trait specified.

Supplemental Table 8: Colocation with Mendelian cardiomyopathy genes

ACTN2	FLNC	NKX2-5	RBM20	TTN
ALPK3	GATA4	PLEKHM2	RYR2	
BAG3	MYH6	PLN	SHOC2	
CSRP3	MYH7	PTPN11	TMEM43	

Each listed gene is located within 500 kilobases of a lead SNP from one of the seven traits, and was found in the list of 129 Mendelian cardiomyopathy-related genes in **Supplemental Table 7**.

Supplemental Table 9: PheWAS results for curated disease phenotypes

Curated Disease Phenotype	PRS	N	N With Phenotype	Beta	SE	-log10(P)
Dilated cardiomyopathy	lvesvi	449027	923	0.413	0.034	33.07
Hypertension	lvedv	449027	156957	-0.039	0.003	31.01
Dilated cardiomyopathy	lvesv	449027	923	0.386	0.034	29.93
Dilated cardiomyopathy	lvef	449027	923	-0.384	0.034	28.01
Hypertension	sv	449027	156957	-0.036	0.003	27.20
Heart_Failure	lvesvi	449027	9226	0.109	0.011	23.52
Hypothyroidism	lvedvi	449027	28703	-0.059	0.006	20.63
Hypothyroidism	sv	449027	28703	-0.059	0.006	20.39
Hypertrophic cardiomyopathy	lvesvi	449027	357	-0.450	0.051	18.15
Hypothyroidism	lvesv	449027	28703	-0.054	0.006	17.52
Hypertrophic cardiomyopathy	lvef	449027	357	0.451	0.052	17.52
Psoriasis	lvedv	449027	7008	-0.100	0.012	16.31
Heart_Failure	lvef	449027	9226	-0.090	0.011	15.96
Psoriasis	sv	449027	7008	-0.097	0.012	15.30
Coronary_Artery_Disease	sv	449027	25195	-0.052	0.007	14.24
Hypertrophic cardiomyopathy	lvesv	449027	357	-0.395	0.053	13.26
Heart_Failure	lvesv	449027	9226	0.077	0.011	12.33
Hypercholesterolemia	sv	449027	85925	-0.029	0.004	12.15
Venous_thromboembolism	svi	449027	16854	0.057	0.008	11.68
Myocardial_Infarction	sv	449027	19286	-0.049	0.008	10.24
Hypertension	lvedvi	449027	156957	-0.021	0.003	9.95
Hypercholesterolemia	lvedv	449027	85925	-0.025	0.004	9.83
Dilated cardiomyopathy	lvedvi	449027	923	0.203	0.033	8.94

Coronary_Artery_Disease	lvedv	449027	25195	-0.041	0.007	8.93
Dilated cardiomyopathy	lvedv	449027	923	0.198	0.033	8.64
Rheumatoid_arthritis	sv	449027	8751	-0.064	0.011	8.35
Pulmonary_embolism	svi	449027	5032	0.085	0.015	8.04
Mitral_valve_disease	lvef	449027	5978	-0.075	0.013	7.79
Myocardial_Infarction	lvedv	449027	19286	-0.042	0.008	7.57
Mitral_regurgitation	lvef	449027	3395	-0.097	0.018	7.34
Hypertension	lvesvi	449027	156957	0.018	0.003	7.11
Atrial_fibrillation_or_flutter	lvesv	449027	21367	-0.038	0.007	7.07
Hypothyroidism	lvedv	449027	28703	-0.032	0.006	6.77
Cardiac_surgery	sv	449027	8053	-0.056	0.011	6.13
Asthma	sv	449027	60959	0.022	0.004	6.01
Mitral_valve_disease	lvesvi	449027	5978	0.064	0.013	5.88
Atrial_fibrillation_or_flutter	lvesvi	449027	21367	-0.034	0.007	5.83
Mitral_regurgitation	lvesvi	449027	3395	0.082	0.017	5.65
Coronary_Artery_Disease	lvef	449027	25195	-0.031	0.007	5.46
Diabetes_Type_1	lvesv	449027	3914	-0.075	0.016	5.44
Hypertension	svi	449027	156957	-0.015	0.003	5.14
Breast_cancer	lvedvi	449027	16179	0.036	0.008	4.93
Myocardial_Infarction	lvef	449027	19286	-0.033	0.008	4.93
Diabetes_Type_2	lvedv	449027	28041	-0.027	0.006	4.82
Implantable_cardioverter_defibrillator	lvesvi	449027	1051	0.132	0.031	4.61
Coronary_Artery_Disease	lvedvi	449027	25195	-0.027	0.007	4.27
Peripheral_vascular_disease	lvesv	449027	6426	-0.051	0.013	4.25
Diabetes_Type_1	sv	449027	3914	-0.065	0.016	4.23
Hernia	lvedv	449027	59636	-0.018	0.004	4.20

Effect size, standard error, and P value are displayed for the association between manually curated phenotypes and the seven cardiac trait polygenic scores. Using Bonferroni correction to account for multiple testing (7 scores, 96 phenotypes), an association $P < 7.4 \times 10^{-5}$ with any of the 7 polygenic scores is considered significant and is listed in this table.

Supplemental Table 13: Imputed TTNtv were not confirmed by exome sequencing

SNP	N Imputed	N Confirmed By Sequence	Cohort-wide MAF	INFO
rs565761937	40	0	8.21E-04	0.66
rs557312035	6	0	9.31E-05	0.45
rs565675340	6	0	8.26E-04	0.25
rs574660186	3	0	5.65E-05	0.95
rs548010682	2	0	4.35E-05	0.77
rs185589320	2	0	5.86E-05	0.86
rs112188483	2	0	3.45E-05	0.35
rs542074139	2	0	2.25E-05	0.77
rs140743001	1	0	1.62E-04	0.27
rs561946873	1	0	4.03E-05	0.40
rs145423907	1	0	6.13E-04	0.20

Genome-wide genotyping data was imputed and provided by the UK Biobank. Within the subset of 49,997 participants with exome sequencing data, 66 had imputed SNPs predicted to create a

TTNtv. However, 0 of these 66 were confirmed in exome sequencing. Each imputed SNP, the number of participants imputed as having the SNP, the MAF of the SNP in the entire UK Biobank, and the imputation INFO score are displayed.

Reviewers' Comments:

Reviewer #1:

Remarks to the Author:

I respectfully disagree with the authors apparent view that because no other P value threshold is widely agreed upon in the field the use of $P < 5 \times 10^{-8}$ is justified. One way to solve this is to state in the paper the number of identified loci with the application of a more stringent P value (e^{-9}) as was done in PMID: 30061737. Stating the number of identified loci applying a less stringent P value is not helpful. I appreciate the authors attempt to replicate their findings and their responses to my other comments.

Additional comments not mentioned in the original review:

It is unclear if the automated cardiac MRI measures are provided by the UK Biobank or performed on the images by the study group, it would be helpful to clarify this point.

Importantly I note that it appears that the authors use the whole UKB database, that is individuals of all ancestries, for their analyses. I think it is doubtful that BOLT suffices to account for the resulting population stratification and ancestral heterogeneity. Although individuals of European ancestry account for 99% of participants with cardiac MRI data, in general individuals of European ancestry have been estimated to account for a little over 80% of the total number of UK Biobank participants. Thus it is conceivable that this affects other analyses, in particular the PRS associations and heritability estimates (which appear high). I strongly recommend that all analyses be performed and reported for British-only also. I apologize for not noting this before.

Reviewer #2:

Remarks to the Author:

I am grateful to the authors for their thoughtful responses to my comments and for their thorough and comprehensive resolution of the issues that I raised. The inclusion of the TWAS analysis and increase in minimum case number for the pheWAS analysis add considerably to the manuscript. I am now satisfied that the manuscript is acceptable.

“Analysis of cardiac magnetic resonance imaging traits in 36,000 individuals reveals shared genetic basis with dilated cardiomyopathy”

Editor's comments:

Your manuscript entitled "Analysis of cardiac magnetic resonance imaging traits in 36,000 individuals reveals shared genetic basis with dilated cardiomyopathy" has now been seen again by 2 referees. You will see from their comments below that while they find your work improved, some important points are still raised. We are interested in the possibility of publishing your study in Nature Communications, but would like to consider your response to these concerns in the form of a revised manuscript before we make a final decision on publication.

We therefore invite you to revise and resubmit your manuscript, taking into account the points raised. Please highlight all changes in the manuscript text file.

We are committed to providing a fair and constructive peer-review process. Do not hesitate to contact us if you wish to discuss the revision in more detail or if there are specific requests from the reviewers that you believe are technically impossible or unlikely to yield a meaningful outcome.

Reviewer #1

I respectfully disagree with the authors' apparent view that because no other P value threshold is widely agreed upon in the field the use of $P < 5 \times 10^{-8}$ is justified. One way to solve this is to state in the paper the number of identified loci with the application of a more stringent P value ($e-9$) as was done in PMID: 30061737. Stating the number of identified loci applying a less stringent P value is not helpful. I appreciate the authors attempt to replicate their findings and their responses to my other comments.

Author response:

We appreciate the careful evaluation by the Reviewer and we believe that our revised manuscript has been significantly improved as a result of the review process.

Based on the Reviewer's thoughtful recommendation and example, we have added a supplementary table that describes the number of significant loci per trait across a range of P value thresholds (**Supplemental Table 3**), and stated in the text the number of loci that would have met criteria for significance with a stricter P value threshold (5×10^{-9}). We have also expanded our discussion and limitations section to address this important point.

Although we acknowledge that others have used more stringent P value thresholds when reporting their primary results, we feel it may be particularly appropriate to emphasize results using a threshold of 5×10^{-8} for the current study.

A key subtext of our study was to compare the relative yield of a population genetic analysis of cardiac imaging traits with that of a population genetic analysis of heart failure disease

phenotypes. Each is a distinct and viable approach for uncovering the genetic determinants of heart failure. Emphasizing the yield of our cardiac MRI GWAS at the established threshold of 5×10^{-8} would permit direct comparison of our results with, for example, those of the HERMES Consortium -- an international, mega-GWAS of a heart failure disease phenotype that reported 11 disease-associated loci at $P < 5 \times 10^{-8}$ as published recently in Nature Communications (Shah *et al.*, 2020, "Genome-wide association and Mendelian randomisation analysis provide insights into the pathogenesis of heart failure."). This comparison of analytic approaches may have implications for future genetic discovery efforts in the field of heart failure.

Manuscript changes:

In the **Results** section, we have updated the text to read as follows, with changes in bold:

Having established a genetic basis for variability in cardiac structure and function, we then performed a series of GWAS to identify common genetic variants associated with the seven cardiac MRI phenotypes. 57 distinct loci in the human genome were associated with at least one cardiac MRI phenotype **at a widely used genome-wide significance threshold** ($P < 5 \times 10^{-8}$; Table 1). **If a more stringent threshold of $P < 5 \times 10^{-9}$ had been used, we would have identified 36 distinct loci (Supplemental Table 3)(Nielsen *et al.*, 2018; Wu *et al.*, 2017).** LD score regression revealed minimal test statistic inflation, consistent with polygenicity rather than population stratification (Supplemental Table 3). No lead SNP deviated from Hardy-Weinberg equilibrium (HWE) beyond the threshold of HWE $P = 1 \times 10^{-6}$.

In the **Discussion** section, we have updated the text to read as follows, with changes in bold:

...Second, genetic analyses of quantitative cardiac imaging traits may improve our understanding of the common genetic basis of cardiomyopathies. In our prior genetic analysis of the UK Biobank, we refined a heterogeneous heart failure phenotype to a specific, nonischemic cardiomyopathy subset, enabling detection of two DCM risk loci (near BAG3 and CLCNKA) that associated with subclinical changes in LV structure and function (Aragam Krishna G *et al.*, 2018). Similarly, seminal GWAS of dilated cardiomyopathy yielded nine risk loci, but were limited by the recruitment of cardiomyopathy cases (Esslinger *et al.*, 2017; Meder *et al.*, 2014; Villard *et al.*, 2011). **A collaborative effort to boost power by aggregating all heart failure subtypes across a number of studies yielded several common genetic loci for heart failure risk factors, i.e. coronary artery disease and atrial fibrillation, but few loci for cardiomyopathies (Shah *et al.*, 2020).** By comparison, recent work analyzing the first 17,000 cardiac MRI studies from the UK Biobank reidentified eight loci previously found to be associated with cardiomyopathy or cardiac imaging traits (Aung Nay *et al.*, 2019). Here, we pursued a genetic analysis including a larger set of cardiac MRI studies from the UK Biobank (total sample size = 36,041)...

....There are several limitations to our findings. First, our analyses were limited to older individuals of predominantly European ancestry, which may limit their applicability to younger individuals and those of other ancestries. Second, the cardiac measurements are derived from automated readings. Third, because the UK Biobank relied on hospitalization or death to assign disease status, unrecognized disease at baseline may

have occurred for individuals without any pre-enrollment hospitalizations. Fourth, the **commonly used** GWAS P value significance threshold of 5×10^{-8} in European populations was derived from Encyclopedia of DNA Elements (ENCODE) data in a small sample (Pe'er et al., 2008); while this threshold remains commonly used, there is no universally agreed upon P value threshold that accounts for larger sample sizes and rarer minor allele frequencies **although more stringent thresholds have been proposed** (Fadista et al., 2016; Wu et al., 2017).

We have also included **Supplemental Table 3** at the bottom of this document.

Additional comments not mentioned in the original review:

It is unclear if the automated cardiac MRI measures are provided by the UK Biobank or performed on the images by the study group, it would be helpful to clarify this point.

Author response:

The automated measures were provided by the UK Biobank. We now clarify this in the text.

Manuscript changes:

In the **Results** section, we have updated the text to read as follows, with changes in bold:

We identified 36,041 UK Biobank participants with cardiac MRI readings **provided by the UK Biobank** who did not have a diagnosis of congestive heart failure (CHF), coronary artery disease (CAD), or DCM at the time of enrollment.

In the **Methods** section, we have updated the text to read as follows, with changes in bold:

Cardiac assessment was performed from the combination of several cine series using balanced steady-state free precession acquisitions, with post-processing by cvi42 Version 5.1.1 (Petersen et al., 2017a). **All measurements were provided by the UK Biobank.** Because of known bias in the vD13A automated measurements, a bias correction was applied for left ventricular end diastolic volume (LVEDV) and left ventricular end systolic volume (LVESV) measurements, using linear corrections derived from a UK cohort undergoing imaging on the same MRI platform (Sanghvi et al., 2016).

Importantly I note that it appears that the authors use the whole UKB database, that is individuals of all ancestries, for their analyses. I think it is doubtful that BOLT suffices to account for the resulting population stratification and ancestral heterogeneity. Although individuals of European ancestry account for 99% of participants with cardiac MRI data, in general individuals of European ancestry have been estimated to account for a little over 80% of the total number of UK Biobank participants. Thus it is conceivable that this affects other analyses, in particular the PRS associations and heritability estimates (which appear high). I strongly recommend that all analyses be performed and reported for British-only also. I apologize for not noting this before.

Author response:

We appreciate that the Reviewer has raised this point of particular importance. We have now attempted to address this point as the Reviewer suggested: by repeating the genetic analyses using only European-ancestry samples.

While 98% of the 36,041 participants in our study were self-identified as European (N=35,407), imposing genetic homogeneity reduces this sample size further. We used the *aberrant* package to identify a group of genetically similar individuals of self-described European ancestry. This is the same procedure used in Bycroft, *et al*, to define the “white British” cohort. It differs in that we also permit Irish and other European nationalities to count as European (previously described in Roselli *et al*, 2018, “Multi-Ethnic Genome-wide Association Study for Atrial Fibrillation,” *Nat Genet*). This procedure reduced the full sample of 36,041 down to 32,755 participants (90.9%) who were considered to be part of the stringently defined genetic “ingroup” of European ancestry.

We then conducted a European-specific analysis on this ingroup: we described the clinical characteristics of these participants, estimated heritability of the 7 cardiac MRI traits, performed GWAS, and conducted polygenic score analyses in this subset.

The heritability estimates using BOLT-REML were all slightly higher in the European subset compared to the main analysis. For example, the heritability estimate of LVEDV increased from 0.426 to 0.445, with full results in **Supplemental Table 16**.

The number of loci with $P < 5 \times 10^{-8}$ decreased from 57 in the main analysis to 49 in the European-specific analysis. 12 loci from the main analysis were no longer significant with $P < 5 \times 10^{-8}$ in the European-specific analysis, while 4 loci newly achieved that P threshold. We observed similar effect estimates at each SNP and nearly perfect effect estimate concordance for all SNPs which achieved $P < 5 \times 10^{-6}$ in the main analysis. No SNP with $P < 5 \times 10^{-8}$ in the main analysis had its direction of effect altered in the European-specific analysis.

The LVESVi polygenic score continued to be the score most strongly associated with dilated cardiomyopathy. We constructed scores using SNP effect-size-weighting from the European-specific analysis, and applied these scores in the remaining unrelated UK Biobank cohort that fit the same strict inlier definition of European ancestry. The hazard ratio per standard deviation change in the new LVESVi polygenic score in these samples was 1.56, similar to that of the LVESVi score produced from the main analysis that was applied in all unrelated samples regardless of ancestry (HR 1.58).

Overall, we interpreted these results as consistent with a slight loss of power from removing about 9% of samples from the study due to not meeting our strict European definition, rather than as supporting an alternative hypothesis that the genetic outlier samples (included in the main analysis) had outsized influence on SNP effect estimates or polygenic disease risk.

Manuscript changes:

In the **Methods** section, we have clarified the LD panel that we used with BOLT:

We performed genome-wide association studies (GWAS) using linear mixed models with BOLT-LMM (version 2.3.2) to account for ancestral heterogeneity, cryptic population structure, and sample relatedness (Loh *et al.*, 2015, 2018). **As BOLT-LMM requires a linkage disequilibrium panel, we used the European linkage disequilibrium panel provided with BOLT.** We conducted a GWAS for the rank-based inverse normal transformed values of each of 7 cardiac MRI phenotypes: LVEDV, LVESV, LVEF, SV, LVEDVi, LVESVi, and SVi. Each GWAS was adjusted for the first five principal components of ancestry, sex, year of birth, age at the time of MRI, and the MRI

scanner's unique identifier to account for batch effects. Variants with association $P < 5 \times 10^{-8}$ were considered to be genome-wide significant.

In the **Results** section, we have added a section on the European-specific analyses that points to supplemental methods and results:

Sensitivity analysis in European-ancestry participants

To understand whether our results were confounded by residual population stratification, the primary genetic analyses (heritability assessment, GWAS, and polygenic score estimates) were repeated in a subset consisting of samples that were within a tight genetic inlier cluster among participants with self-reported European ancestry ($N = 32,755$). These sensitivity analyses yielded similar heritability estimates, fewer genome-wide significant loci (49 instead of 57), and similar polygenic score effect estimates when compared to the main analyses using all 36,041 samples (see **Supplemental Methods, Supplemental Results, Supplemental Figure 8, and Supplemental Tables 15-24**).

We have added the following to the **Supplemental Methods** section:

Sensitivity analysis in Europeans

Starting with participants in the UK Biobank who self-identified as British, Irish, or Other European, we applied the aberrant R package as previously described (Bycroft et al., 2018; Roselli et al., 2018). Briefly, we used the first 6 principal components (PCs) of ancestry in pairs (PC 1 and 2; PC 3 and 4; PC 5 and 6) with lambda (the ratio of standard deviations of outliers vs inliers) set to 40. We took the intersection of these 3 pairs, isolating a tight genetic inlier cluster of European participants.

For each trait, we then conducted a heritability analysis with BOLT-REML and a GWAS with BOLT-LMM. Using the lead SNPs from this smaller European sample, we produced polygenic scores and applied them in the European subset of the remaining unrelated UK Biobank participants who had not undergone cardiac MRI. Aside from using this genetically defined subset of European participants, all analyses were conducted in the same fashion and with the same settings as in the main analysis.

To evaluate the SNP-level differences in the GWAS, we compared effect estimates (Beta) for each SNP in the main analysis with those in the European-specific analysis using a linear model. Taking SNPs below a given P value threshold in the main analysis, agreement was assessed by the overall model fit (r^2). This procedure was performed for each trait across several P value thresholds.

We have added a **Supplemental Results** section:

Sensitivity analysis in Europeans

Because we ran BOLT with the default European linkage disequilibrium panel for the main analyses, we also conducted sensitivity analyses using a subset of rigorously-defined European samples to understand whether our results were influenced by residual population stratification. To do this, we defined a genetically similar subset of participants who self-described as British, Irish, or another European ancestry and underwent cardiac MRI ($N = 32,755$, 9% fewer samples than in the main analysis). The characteristics of these participants are described in **Supplemental Table 15**.

The heritability estimates for each cardiac trait in the European-specific analysis were similar to those in the main analysis (**Supplemental Table 16**). The GWAS in Europeans showed similar evidence for polygenicity, and lack of confounding, as were observed in the main analysis (**Supplemental Table 17**).

Eight fewer loci achieved genome-wide significance in the European-specific analysis (49 distinct loci instead of 57 in the main analysis, or fewer loci with more stringent P value thresholds as in **Supplemental Table 18**). Four loci newly achieved genome-wide significance in the European-specific analysis (**Supplemental Table 19**). These include loci near *ARPC1A*, whose gene product is actin related protein 2/3 complex subunit 1A; *PDGFD*, whose gene product is platelet derived growth factor D; *LSM7*; and *LYN*. Twelve of the lead SNPs identified in the main analysis with $P < 5 \times 10^{-8}$ did not have $P < 5 \times 10^{-8}$ in the European-specific analysis for any trait, although all SNPs with $P < 5 \times 10^{-8}$ in the main analysis had association $P < 3 \times 10^{-6}$ or stronger in the European-specific analysis. All lead SNPs from the main analysis are shown with their corresponding European-specific effect size and P value in **Supplemental Table 20**.

Across various P value thresholds, the effect estimates for each SNP showed strong correlation with those in the main analysis, with nearly perfect correlation for all SNPs beyond $P < 5 \times 10^{-6}$ (**Supplemental Table 21, Supplemental Figure 8**). Of the SNPs with $P < 5 \times 10^{-8}$ in the main analysis, none had an effect in the opposite direction in the European-specific analysis (**Supplemental Table 22**).

Polygenic scores produced using the same procedure as in the main analysis yielded similar hazard ratios (participant characteristics in **Supplemental Table 23** and results in **Supplemental Table 24**). The best performing score, derived from the LVESVi lead SNPs, yielded an HR of 1.56 for incident dilated cardiomyopathy in the European-only subset, similar to the HR of 1.58 from the polygenic score produced in the main analysis.

We have also included **Supplemental Tables 15-24** and **Supplemental Figure 8** at the bottom of this document.

Reviewer #2

I am grateful to the authors for their thoughtful responses to my comments and for their thorough and comprehensive resolution of the issues that I raised. The inclusion of the TWAS analysis and increase in minimum case number for the pheWAS analysis add considerably to the manuscript. I am now satisfied that the manuscript is acceptable.

Author response:

We appreciate the Reviewer's thoughtful feedback and believe that their questions and suggestions have significantly improved the manuscript.

Modified Tables and Figures

Supplemental Table 3: Loci significant at various P value thresholds

P threshold	LVEDV	LVEDVi	LVESV	LVESVi	LVEF	SV	SVi	Total distinct loci
5.00E-08	22	14	22	32	28	12	8	57
1.00E-08	17	7	15	23	21	9	5	37
5.00E-09	17	7	15	21	19	9	3	36
1.00E-09	14	5	14	17	15	5	2	28

“P threshold” represents the P value cutoff. “Total distinct loci” is a count of unique loci aggregated across all traits, so that any locus found in two or more traits is only counted once.

Supplemental Table 15: Clinical characteristics of European UK Biobank participants with cardiac MRI data

		Women	Men	All
N		17318	15437	32755
Age at MRI				
	Mean (SD)	64 (7.4)	65 (7.6)	64 (7.5)
	Median [Q1-Q3]	64 [58 - 69]	66 [59 - 71]	65 [58 - 70]
Ancestry				
	European	17318 (100 %)	15437 (100 %)	32755 (100 %)
BMI (kg/m²)				
	Mean (SD)	26 (4.7)	27 (3.9)	26 (4.4)
	Median [Q1-Q3]	25 [23 - 28]	26 [24 - 29]	26 [23 - 29]
	Missing	497 (2.9%)	386 (2.5%)	883 (2.7%)
Height (cm)				
	Mean (SD)	160 (6.2)	180 (6.6)	170 (9.3)
	Median [Q1-Q3]	160 [160 - 170]	180 [170 - 180]	170 [160 - 180]
	Missing	450 (2.6%)	350 (2.3%)	800 (2.4%)
Weight (kg)				
	Mean (SD)	69 (13)	84 (13)	76 (15)
	Median [Q1-Q3]	67 [60 - 76]	82 [75 - 91]	75 [65 - 85]
	Missing	454 (2.6%)	372 (2.4%)	826 (2.5%)
SBP (mmHg)				
	Mean (SD)	130 (18)	140 (16)	140 (18)
	Median [Q1-Q3]	130 [120 - 140]	140 [130 - 150]	130 [120 - 150]
	Missing	7 (0.0%)	0 (0%)	7 (0.0%)

DBP (mmHg)				
	Mean (SD)	79 (9.7)	84 (9.6)	81 (9.9)
	Median [Q1-Q3]	79 [73 - 86]	84 [77 - 90]	81 [75 - 88]
	Missing	7 (0.0%)	0 (0%)	7 (0.0%)
MET minutes/week				
	Mean (SD)	2400 (3000)	2600 (3400)	2500 (3200)
	Median [Q1-Q3]	1500 [620 - 2900]	1600 [690 - 3200]	1500 [660 - 3100]
Standard drinks/week				
	Mean (SD)	9.2 (6.9)	15 (11)	12 (9.4)
	Median [Q1-Q3]	7.3 [4.5 - 12]	12 [7.3 - 19]	9.3 [6.0 - 16]
	Missing	4978 (28.7%)	2563 (16.6%)	7541 (23.0%)
Pack year smoking history				
	Mean (SD)	3.6 (9.0)	5.7 (13)	4.6 (11)
	Median [Q1-Q3]	0.0 [0.0 - 0.0]	0.0 [0.0 - 3.8]	0.0 [0.0 - 0.0]
Smoking status at enrollment				
	Current	861 (5 %)	1090 (7 %)	1951 (6 %)
	Never	11136 (64 %)	8847 (57 %)	19983 (61 %)
	Prefer_not_to_answer	34 (0 %)	30 (0 %)	64 (0 %)
	Previous	5287 (31 %)	5470 (35 %)	10757 (33 %)
LVEDV				
	Mean (SD)	120 (20)	150 (28)	140 (29)
	Median [Q1-Q3]	120 [110 - 140]	150 [130 - 170]	130 [120 - 150]
LVEDVi				
	Mean (SD)	70 (11)	76 (14)	73 (12)
	Median [Q1-Q3]	70 [63 - 77]	76 [67 - 85]	72 [65 - 80]
	Missing	480 (2.8%)	381 (2.5%)	861 (2.6%)
LVESV				
	Mean (SD)	42 (11)	58 (17)	49 (16)
	Median [Q1-Q3]	41 [34 - 48]	56 [47 - 67]	47 [38 - 58]
LVESVi				
	Mean (SD)	24 (6.1)	29 (8.0)	26 (7.5)
	Median [Q1-Q3]	23 [20 - 27]	28 [24 - 33]	25 [21 - 30]
	Missing	480 (2.8%)	381 (2.5%)	861 (2.6%)
LVEF				
	Mean (SD)	0.67 (0.052)	0.63 (0.058)	0.65 (0.058)
	Median [Q1-Q3]	0.67 [0.64 - 0.70]	0.63 [0.59 - 0.66]	0.65 [0.61 - 0.69]
SV				
	Mean (SD)	81 (12)	96 (17)	88 (16)
	Median [Q1-Q3]	81 [73 - 89]	95 [85 - 110]	87 [77 - 98]

SVi				
	Mean (SD)	46 (6.6)	47 (8.1)	47 (7.4)
	Median [Q1-Q3]	46 [42 - 51]	47 [42 - 53]	47 [42 - 52]
	Missing	480 (2.8%)	381 (2.5%)	861 (2.6%)

Supplemental Table 16: Heritability in the full sample and in the European-specific subset

	Heritability	
	All participants	European participants
LVEDV	0.426	0.445
LVESV	0.400	0.417
SV	0.342	0.364
LVEF	0.313	0.332

SNP heritability was assessed with BOLT-REML. A sensitivity analysis that excluded all outliers from the genetically European subset revealed similar heritability estimates to those from the main analyses.

Supplemental Table 17: LD score regression in the European-specific analysis

	Lambda	Intercept
LVEDV	1.1459	1.0258
LVEDVi	1.1459	1.0047
LVESV	1.1459	1.0088
LVESVi	1.1459	0.9973
LVEF	1.0957	0.9901
SV	1.1459	1.0277
SVi	1.0957	1.0092

Lambda represents the genomic inflation factor. Intercept represents the intercept from LD score regression.

Supplemental Table 18: Loci significant at various P value thresholds in the European-specific analysis

P threshold	LVEDV	LVEDVi	LVESV	LVESVi	LVEF	SV	SVi	Total distinct loci
5.00E-08	19	10	19	28	21	11	7	49

1.00E-08	15	5	16	16	15	6	4	28
5.00E-09	13	5	14	15	15	5	3	26
1.00E-09	10	5	11	14	12	4	2	19

Supplemental Table 19: Loci newly achieving genome-wide significance in the European-specific analysis

Trait	SNP	CHR	BP	A1	A0	A1 Freq	A1 Freq Euro	INFO	BETA	BETA Euro	P	P Euro	Nearest Gene	Novel in Study
lvedv	rs71517989	8	56877592	A	AT	0.554	0.554	0.99	0.0314	0.0367	1.60E-07	7.70E-09	LYN	Yes
lvef	rs36234	16	2201270	G	C	0.637	0.635	0.91	0.0399	0.0426	6.90E-08	3.00E-08	RAB26	No
lvesv	rs113781447	19	2317945	T	A	0.982	0.982	0.92	-0.1315	-0.1419	1.20E-07	3.20E-08	LSM7	Yes
lvesv	rs377751939	5	138734734	T	C	0.807	0.814	0.84	0.0456	0.0495	6.00E-08	2.50E-08	SPATA24	No
sv	rs149613324	11	104215908	G	C	0.991	0.991	0.84	-0.1827	-0.2070	2.80E-07	2.40E-08	PDGFD	Yes
svi	rs10258757	7	98970675	C	T	0.863	0.872	1.00	-0.0586	-0.0663	7.70E-08	7.70E-09	ARPC1A	Yes

BP is the base pair distance, keyed to GRCh37. A1 is the effect allele, A0 is the non-effect allele. A1 Freq Euro represents the effect allele frequency in the European-specific analysis. BETA Euro and P Euro, respectively, represent the effect size and P value in the European-specific analysis. Novel in Study indicates whether a locus was detected for another trait in the main analysis (“No”) or whether the locus was not detected in the main analysis (“Yes”).

Supplemental Table 20: Comparison of all genome-wide significant loci from the main analysis in the European-specific analysis

Trait	SNP	CHR	BP	A1	A0	A1 Freq	A1 Freq Euro	BETA	BETA Euro	P	P Euro	Locus ID	Nearest Gene
lvedv	rs28579893	1	16347534	A	G	0.327	0.328	-0.0369	-0.0368	1.00E-09	3.50E-09	1	CLCNKA
lvedv	rs753562515	1	46007032	CAA	C	0.561	0.563	-0.0353	-0.0352	4.40E-09	6.90E-08	2	AKR1A1
lvedv	rs7605066	2	71529331	C	T	0.429	0.426	0.0332	0.0339	1.60E-08	3.00E-08	3	ZNF638
lvedv	rs539762056	2	174892676	A	AT	0.649	0.648	0.0352	0.0376	3.40E-08	1.60E-08	4	SP3
lvedv	rs1873164	2	179753549	G	A	0.200	0.195	-0.0604	-0.0633	1.20E-16	1.50E-16	5	CCDC141
lvedv	rs73028849	3	14272766	G	C	0.658	0.657	0.0396	0.0377	2.10E-10	7.90E-09	6	XPC
lvedv	rs6777123	3	169303070	A	C	0.393	0.393	0.0335	0.0334	1.80E-08	8.10E-08	7	MECOM
lvedv	rs4521636	6	31312293	T	C	0.466	0.477	0.0390	0.0400	1.10E-10	1.80E-10	8	HLA-B
lvedv	rs9275587	6	32680379	T	C	0.525	0.533	0.0370	0.0382	8.90E-10	1.20E-09	9	HLA-DQA2
lvedv	rs2146324	6	43756863	A	C	0.261	0.266	0.0386	0.0379	1.00E-08	7.80E-08	10	VEGFA
lvedv	rs11153730	6	118667522	T	C	0.513	0.507	-0.0482	-0.0477	3.70E-16	2.00E-14	11	PLN
lvedv	rs3918226	7	150690176	C	T	0.921	0.919	0.0585	0.0566	4.90E-08	4.70E-07	12	NOS3
lvedv	8:125858538_GA_G	8	125858538	GA	G	0.688	0.686	0.0422	0.0421	1.80E-11	2.90E-10	13	MTSS1
lvedv	rs72840788	10	121415685	G	A	0.785	0.782	0.0549	0.0538	1.50E-14	8.30E-13	14	BAG3
lvedv	rs7306710	12	66376091	T	C	0.480	0.487	0.0391	0.0373	5.90E-11	1.20E-09	15	LLPH
lvedv	rs3184504	12	111884608	T	C	0.470	0.478	-0.0477	-0.0495	3.30E-16	7.80E-16	16	SH2B3
lvedv	rs10850034	12	112817521	T	A	0.647	0.643	0.0504	0.0535	7.60E-15	4.00E-15	17	HECTD4
lvedv	15:85348961_TTTTG_T	15	85348961	TTTTG	T	0.746	0.744	-0.0400	-0.0443	2.80E-09	3.50E-10	18	ZNF592
lvedv	rs71385734	16	2160503	T	G	0.828	0.829	0.0504	0.0563	2.10E-10	1.90E-11	19	PKD1
lvedv	rs2302455	17	1374195	G	A	0.884	0.882	0.0591	0.0546	1.30E-10	1.20E-08	20	MYO1C
lvedv	19:41945122_CT_C	19	41945122	CT	C	0.606	0.607	-0.0371	-0.0345	6.80E-10	3.70E-08	21	ATP5SL
lvedv	rs12460541	19	46312077	G	A	0.653	0.652	-0.0398	-0.0399	6.90E-11	2.90E-10	22	RSPH6A
lvedvi	rs1976402	1	16143779	G	A	0.712	0.709	0.0435	0.0432	1.10E-08	5.10E-08	1	SPEN
lvedvi	rs1873164	2	179753549	G	A	0.200	0.196	-0.0701	-0.0742	1.80E-15	1.40E-15	5	CCDC141

lvedvi	rs73028849	3	14272766G	C	0.659	0.658	0.0439	0.0417	4.10E-09	1.10E-07	6XPC
lvedvi	rs6458349	6	43759789G	A	0.272	0.277	0.0444	0.0455	1.50E-08	3.20E-08	10VEGFA
lvedvi	rs72967533	6	118655020T	C	0.521	0.516	-0.0488	-0.0484	7.10E-12	1.20E-10	11PLN
lvedvi	8:125858538 _GA_G	8	125858538GA	G	0.688	0.687	0.0455	0.0453	2.30E-09	1.50E-08	13MTSS1
lvedvi	rs72840788	10	121415685G	A	0.785	0.782	0.0588	0.0571	1.00E-11	2.50E-10	14BAG3
lvedvi	rs35350651	12	111907431A	AC	0.483	0.492	-0.0441	-0.0448	2.40E-10	6.90E-10	16ATXN2
lvedvi	rs10850034	12	112817521T	A	0.648	0.643	0.0424	0.0443	3.90E-08	4.20E-08	17HECTD4
lvedvi	rs7502466	17	1372970G	A	0.891	0.889	0.0634	0.0570	2.30E-08	1.50E-06	20MYO1C
lvedvi	rs9797817	19	46312345C	T	0.653	0.651	-0.0508	-0.0512	4.60E-12	3.90E-11	22RSPH6A
lvedvi	rs767987273	2	201170509CA	C	0.602	0.602	0.0399	0.0401	4.80E-08	2.20E-07	23SPATS2L
lvedvi	rs12499670	4	174621493T	C	0.396	0.397	0.0401	0.0360	3.60E-08	3.00E-06	24HAND2
lvedvi	rs9480737	6	107442277A	G	0.683	0.680	-0.0427	-0.0448	2.30E-08	2.20E-08	25BEND3
lvef	rs1739837	1	16337933C	T	0.411	0.413	0.0708	0.0670	2.90E-25	2.60E-21	1HSPB7
lvef	rs2562845	2	179514433T	C	0.803	0.808	-0.0792	-0.0806	5.70E-20	6.60E-19	5TTN
lvef	rs11710541	3	14291679T	C	0.658	0.658	-0.0651	-0.0644	9.10E-20	1.20E-17	6XPC
lvef	rs9274626	6	32636040T	C	0.319	0.324	-0.0417	-0.0433	2.70E-08	2.30E-08	9HLA-DQB1
lvef	rs72840788	10	121415685G	A	0.785	0.782	-0.0997	-0.0977	4.00E-32	1.10E-28	14BAG3
lvef	rs8023658	15	85323220G	T	0.507	0.510	-0.0464	-0.0466	4.60E-11	3.00E-10	18ZNF592
lvef	rs2503715	1	2144107A	G	0.128	0.129	-0.0584	-0.0575	4.00E-08	2.30E-07	26C1orf86
lvef	rs10925197	1	236842077C	G	0.462	0.455	0.0420	0.0443	6.90E-10	5.00E-10	27ACTN2
lvef	rs56099248	3	69857773C	T	0.805	0.802	-0.0607	-0.0591	5.10E-12	9.50E-11	28MITF
lvef	rs35999985	5	138756825A	G	0.299	0.289	0.0480	0.0482	3.80E-10	1.40E-09	29DNAJC18
lvef	rs3176326	6	36647289G	A	0.800	0.803	-0.0777	-0.0748	1.80E-19	6.10E-17	30CDKN1A
lvef	rs3807309	7	128472138G	A	0.886	0.888	-0.0835	-0.0845	4.10E-15	6.70E-14	31FLNC
lvef	rs36029352	8	11786925C	T	0.455	0.452	-0.0380	-0.0394	1.30E-08	2.80E-08	32DEFB136
lvef	rs4073554	8	141704232T	C	0.481	0.480	0.0428	0.0435	1.10E-09	1.40E-09	33PTK2
lvef	rs189569984	10	112544125C	T	0.992	0.991	-0.2009	-0.2081	3.40E-08	5.50E-08	34RBM20
lvef	rs721067	11	19212726T	A	0.918	0.919	0.0697	0.0690	3.60E-08	1.70E-07	35CSRP3
lvef	rs113819537	12	26348429C	G	0.749	0.749	0.0449	0.0451	1.70E-08	5.10E-08	36SSPN
lvef	rs5029142	16	988070T	A	0.623	0.628	-0.0430	-0.0423	3.90E-10	2.50E-09	37LMF1
lvef	rs12452367	17	53374610T	C	0.715	0.713	-0.0550	-0.0538	1.30E-13	3.80E-12	38HLF
lvef	rs2047273	18	34184859T	C	0.678	0.681	-0.0418	-0.0432	3.10E-08	6.50E-08	39FHOD3
lvef	rs10871753	18	55956865G	T	0.489	0.489	-0.0431	-0.0420	6.10E-10	7.30E-09	40NEDD4L
lvef	rs2070458	22	24159307A	T	0.200	0.192	0.0629	0.0642	2.50E-13	3.90E-12	41SMARCB1
lvesv	rs1048302	1	16340879T	G	0.327	0.328	-0.0605	-0.0583	4.40E-22	3.20E-19	1HSPB7
lvesv	rs753562515	1	46007032CAA	C	0.561	0.563	-0.0365	-0.0363	3.40E-09	2.30E-08	2AKR1A1
lvesv	rs2562845	2	179514433T	C	0.803	0.808	0.0770	0.0786	1.30E-23	1.30E-22	5TTN
lvesv	rs11710541	3	14291679T	C	0.658	0.658	0.0583	0.0566	4.90E-20	1.90E-17	6XPC
lvesv	rs9274626	6	32636040T	C	0.319	0.324	0.0393	0.0408	2.90E-09	4.30E-09	9HLA-DQB1
lvesv	rs11153730	6	118667522T	C	0.513	0.507	-0.0375	-0.0366	6.80E-10	1.00E-08	11PLN
lvesv	8:125858538 _GA_G	8	125858538GA	G	0.688	0.686	0.0483	0.0476	2.40E-13	6.10E-12	13MTSS1
lvesv	rs72840788	10	121415685G	A	0.785	0.782	0.0866	0.0847	8.10E-32	3.00E-28	14BAG3
lvesv	rs3184504	12	111884608T	C	0.470	0.478	-0.0388	-0.0402	1.10E-10	1.30E-10	16SH2B3
lvesv	rs10850034	12	112817521T	A	0.647	0.643	0.0411	0.0444	7.60E-10	1.70E-10	17HECTD4
lvesv	15:85348961 _TTTTG_T	15	85348961TTTTG	T	0.746	0.744	-0.0515	-0.0558	1.30E-13	4.20E-14	18ZNF592
lvesv	rs71385734	16	2160503T	G	0.828	0.829	0.0531	0.0589	4.50E-11	6.70E-12	19PKD1
lvesv	rs2302455	17	1374195G	A	0.884	0.882	0.0635	0.0579	8.60E-11	1.10E-08	20MYO1C
lvesv	rs16975238	19	41944985T	A	0.602	0.604	-0.0338	-0.0321	2.00E-08	4.60E-07	21ATP5SL
lvesv	rs10421891	19	46315809A	G	0.645	0.645	-0.0451	-0.0448	6.70E-13	9.70E-12	22RSPH6A
lvesv	rs730506	6	36645968G	C	0.800	0.803	0.0532	0.0506	2.10E-12	1.90E-10	30CDKN1A
lvesv	rs34373805	7	128486363C	T	0.838	0.843	0.0554	0.0566	9.20E-12	4.70E-11	31FLNC
lvesv	rs1962104	8	141635329T	C	0.449	0.446	-0.0371	-0.0374	2.20E-09	6.80E-09	33AGO2
lvesv	rs11604807	11	19231167T	C	0.862	0.862	-0.0493	-0.0510	3.50E-08	4.70E-08	35CSRP3
lvesv	rs113819537	12	26348429C	G	0.749	0.749	-0.0395	-0.0387	1.30E-08	7.20E-08	36SSPN
lvesv	rs3829491	16	1004834T	C	0.613	0.618	0.0344	0.0339	9.60E-09	6.40E-08	37LMF1
lvesv	rs12452367	17	53374610T	C	0.715	0.713	0.0473	0.0463	5.60E-13	2.20E-11	38HLF
lvesv	rs10871753	18	55956865G	T	0.489	0.489	0.0330	0.0347	3.00E-08	2.60E-08	40NEDD4L
lvesv	rs5760061	22	24178279G	A	0.201	0.192	-0.0550	-0.0543	1.30E-13	8.60E-12	41DERL3
lvesv	rs114300540	1	6248182C	T	0.875	0.874	-0.0590	-0.0555	3.20E-10	1.70E-08	42RLD22
lvesv	rs13092177	3	134455794G	T	0.850	0.851	0.0481	0.0457	1.30E-08	1.90E-07	43EPHB1
lvesv	rs1499813	3	171760427T	C	0.591	0.588	0.0351	0.0362	6.90E-09	1.20E-08	44FNDC3B
lvesv	rs10832164	11	14048480C	T	0.486	0.484	-0.0333	-0.0318	3.80E-08	6.70E-07	45RRAS2

lvesv	rs116904997	12	120668534	G	A	0.978	0.977	-0.1125	-0.1137	3.40E-08	1.00E-07	46PXN
lvesv	rs242562	17	44026739	G	A	0.619	0.617	-0.0373	-0.0370	3.90E-09	1.50E-08	47MAPT
lvesv	rs9897002	17	64286494	A	G	0.571	0.569	0.0327	0.0327	4.00E-08	1.80E-07	48PRKCA
lvesv	19:10765478 CG_C	19	10765478	CG	C	0.236	0.238	0.0408	0.0409	1.20E-08	4.00E-08	49ILF3
lvesvi	rs945425	1	16348412	T	C	0.324	0.325	-0.0701	-0.0685	8.50E-23	1.10E-20	1CLCNKA
lvesvi	rs753562515	1	46007032	CAA	C	0.561	0.563	-0.0384	-0.0385	1.70E-08	6.90E-08	2AKR1A1
lvesvi	rs2562845	2	179514433	T	C	0.804	0.809	0.0849	0.0861	1.60E-22	2.40E-21	5TTN
lvesvi	rs73028849	3	14272766	G	C	0.659	0.658	0.0637	0.0619	6.20E-19	9.20E-17	6XPC
lvesvi	8:125858538 GA_G	8	125858538	GA	G	0.688	0.687	0.0507	0.0504	3.30E-12	7.70E-11	13MTSS1
lvesvi	rs72840788	10	121415685	G	A	0.785	0.782	0.0926	0.0908	1.70E-28	2.20E-25	14BAG3
lvesvi	15:85348961 TTTTG_T	15	85348961	TTTTG	T	0.746	0.744	-0.0543	-0.0592	1.40E-11	6.30E-12	18ZNF592
lvesvi	rs2302455	17	1374195	G	A	0.884	0.881	0.0646	0.0579	4.60E-09	4.90E-07	20MYO1C
lvesvi	rs10421891	19	46315809	A	G	0.646	0.645	-0.0528	-0.0525	1.20E-13	2.30E-12	22RSPH6A
lvesvi	rs774290282	2	201198623	CATT	C	0.608	0.608	-0.0377	-0.0370	4.70E-08	1.00E-06	23SPATS2L
lvesvi	rs79502300	3	69856753	C	T	0.805	0.802	0.0537	0.0542	4.40E-10	2.40E-09	28MITF
lvesvi	rs11748963	5	138730037	T	C	0.726	0.730	0.0439	0.0457	5.90E-09	1.10E-08	29PROB1
lvesvi	rs3176326	6	36647289	G	A	0.800	0.803	0.0650	0.0640	1.80E-14	4.30E-13	30CDKN1A
lvesvi	rs34373805	7	128486363	C	T	0.838	0.843	0.0672	0.0660	2.70E-13	2.10E-11	31FLNC
lvesvi	rs1962104	8	141635329	T	C	0.449	0.446	-0.0427	-0.0434	6.40E-10	2.10E-09	33AGO2
lvesvi	rs189569984	10	112544125	C	T	0.991	0.991	0.2142	0.2193	2.10E-09	1.20E-08	34RBM20
lvesvi	rs8063213	16	992961	T	G	0.618	0.622	0.0449	0.0452	7.60E-11	2.60E-10	37LMF1
lvesvi	rs12452367	17	53374610	T	C	0.715	0.713	0.0548	0.0541	1.80E-13	4.10E-12	38HLF
lvesvi	rs10871753	18	55956865	G	T	0.489	0.489	0.0394	0.0421	6.00E-09	1.60E-09	40NEDD4L
lvesvi	rs5760061	22	24178279	G	A	0.201	0.192	-0.0645	-0.0650	7.50E-15	2.90E-13	41DERL3
lvesvi	rs709208	1	6272137	A	G	0.679	0.681	-0.0412	-0.0399	1.90E-08	1.70E-07	42RNF207
lvesvi	rs1499813	3	171760427	T	C	0.591	0.589	0.0398	0.0407	2.80E-09	1.00E-08	44FNDC3B
lvesvi	rs11023059	11	14064392	A	G	0.519	0.517	0.0416	0.0404	6.50E-10	1.50E-08	45RRAS2
lvesvi	rs116904997	12	120668534	G	A	0.978	0.977	-0.1316	-0.1308	1.10E-08	7.30E-08	46PXN
lvesvi	rs242562	17	44026739	G	A	0.619	0.617	-0.0397	-0.0410	3.00E-08	4.20E-08	47MAPT
lvesvi	rs9892651	17	64303793	C	T	0.418	0.416	0.0398	0.0387	2.10E-09	2.30E-08	48PRKCA
lvesvi	rs190093681	2	180094352	C	T	0.997	0.997	0.3742	0.3589	1.60E-08	4.00E-07	50SESTD1
lvesvi	rs2886037	3	158306414	G	A	0.494	0.497	-0.0363	-0.0359	2.90E-08	1.10E-07	51MLF1
sv	rs7573293	2	179753245	C	T	0.275	0.268	-0.0492	-0.0528	2.00E-12	7.70E-13	5CCDC141
sv	rs111721712	6	31315407	C	CT	0.527	0.535	0.0376	0.0376	3.40E-09	1.80E-08	8HLA-B
sv	rs28391274	6	32623786	A	G	0.797	0.794	0.0474	0.0481	1.40E-08	3.40E-08	9HLA-DQB1
sv	rs2146324	6	43756863	A	C	0.261	0.266	0.0411	0.0415	1.10E-08	3.30E-08	10VEGFA
sv	rs72967533	6	118655020	T	C	0.521	0.516	-0.0511	-0.0512	1.40E-15	1.60E-14	11PLN
sv	rs10400419	12	66389968	T	C	0.448	0.451	0.0402	0.0389	5.40E-10	8.30E-09	15LLPH
sv	rs11065979	12	112059557	C	T	0.572	0.565	0.0483	0.0503	3.90E-14	3.00E-14	16ATXN2
sv	rs11066188	12	112610714	G	A	0.602	0.597	0.0450	0.0466	2.70E-12	3.40E-12	17HECTD4
sv	rs2891403	12	113137572	A	G	0.278	0.281	-0.0418	-0.0423	3.80E-09	9.10E-09	17RPH3A
sv	rs888690	5	172636130	T	C	0.397	0.405	-0.0394	-0.0404	1.20E-09	2.60E-09	52NKX2-5
sv	rs422068	14	23864804	T	C	0.642	0.643	0.0395	0.0391	1.40E-09	1.20E-08	53MYH6
sv	rs143384	20	34025756	A	G	0.588	0.595	-0.0349	-0.0322	3.60E-08	1.00E-06	54GDF5
svi	rs7573293	2	179753245	C	T	0.275	0.268	-0.0506	-0.0547	7.10E-10	2.20E-10	5CCDC141
svi	rs6458349	6	43759789	G	A	0.272	0.277	0.0451	0.0473	3.50E-08	3.40E-08	10VEGFA
svi	rs72967533	6	118655020	T	C	0.521	0.516	-0.0485	-0.0486	1.10E-10	3.70E-10	11PLN
svi	rs9480737	6	107442277	A	G	0.683	0.680	-0.0474	-0.0498	2.60E-09	1.60E-09	25BEND3
svi	rs376439	14	23869029	A	G	0.605	0.607	0.0429	0.0413	6.50E-09	1.10E-07	53MYH6
svi	2:232288831 TTTC_T	2	232288831	TTTC	T	0.684	0.687	-0.0455	-0.0476	9.30E-09	1.00E-08	55B3GNT7
svi	rs1919865	6	122121005	A	T	0.864	0.868	0.0599	0.0595	1.90E-08	1.60E-07	56GJA1
svi	rs579459	9	136154168	T	C	0.794	0.794	-0.0500	-0.0519	4.10E-08	3.60E-08	57SURF6

This table contains the same loci as in **Table 1**. In addition, the lead SNP effect estimates and P values from the European-specific analysis are provided for direct comparison. BP is the base pair distance, keyed to GRCh37. A1 is the effect allele, A0 is the non-effect allele. A1 Freq Euro represents the effect allele frequency in the European-specific analysis. BETA Euro and P Euro, respectively, represent the effect size and P value in the European-specific analysis.

Supplemental Table 21: Concordance of effect estimates from lead SNPs in the main analysis and the European-specific analysis

Trait	P Threshold	Evaluated SNPs	R ²
Ivedv	1	14457773	0.52
	0.05	910097	0.91
	0.005	130905	0.96
	0.0005	26359	0.99
	5.00E-05	8183	0.99
	5.00E-06	3414	1.00
	5.00E-07	1869	1.00
	5.00E-08	1150	1.00
	5.00E-09	552	1.00
Ivedvi	1	14459489	0.52
	0.05	867566	0.91
	0.005	115475	0.97
	0.0005	19745	0.99
	5.00E-05	5466	1.00
	5.00E-06	1690	1.00
	5.00E-07	832	1.00
	5.00E-08	513	1.00
	5.00E-09	256	1.00
Ivesv	1	14458151	0.52
	0.05	907979	0.89
	0.005	134774	0.96
	0.0005	28470	0.99
	5.00E-05	9758	1.00
	5.00E-06	4459	1.00
	5.00E-07	2743	1.00
	5.00E-08	1788	1.00

	5.00E-09	1159	1.00
lvesvi	1	14459698	0.52
	0.05	883056	0.89
	0.005	122307	0.95
	0.0005	23158	0.99
	5.00E-05	7774	0.99
	5.00E-06	3434	1.00
	5.00E-07	2322	1.00
	5.00E-08	1728	1.00
	5.00E-09	1292	1.00
lvef	1	14457483	0.51
	0.05	859758	0.88
	0.005	119329	0.95
	0.0005	24647	0.98
	5.00E-05	7924	1.00
	5.00E-06	4284	1.00
	5.00E-07	2266	1.00
	5.00E-08	1630	1.00
	5.00E-09	1166	1.00
sv	1	14457631	0.52
	0.05	859053	0.92
	0.005	115812	0.97
	0.0005	21115	0.99
	5.00E-05	4696	0.99
	5.00E-06	1754	1.00
	5.00E-07	755	1.00
	5.00E-08	400	1.00
	5.00E-09	216	1.00
svi	1	14458828	0.52
	0.05	828266	0.91
	0.005	105162	0.96

	0.0005	15729	0.99
	5.00E-05	3262	0.99
	5.00E-06	844	1.00
	5.00E-07	285	1.00
	5.00E-08	91	1.00
	5.00E-09	8	1.00

For each trait, effect estimates (Beta) for each SNP in the main analysis were compared with those in the European-specific analysis in a linear model. Using different subsets of SNPs based on the P value in the main analysis, agreement was assessed by the overall model fit (r^2).

Supplemental Table 22: Directional concordance of genome-wide significant SNPs from the main analysis and the European-specific analysis

Trait	Evaluated SNPs	SNPs at P < 5E-08 in the main analysis	Sign mismatches in the European-specific analysis
lvedv	14528119	1150	0
lvedvi	14529827	513	0
lvesv	14528119	1788	0
lvesvi	14529827	1728	0
lvef	14528119	1630	0
sv	14528119	400	0
svi	14529827	91	0

No lead SNPs with P < 5e-8 in the main analysis had an opposite direction of effect in the European-specific analysis.

Supplemental Table 23: Characteristics of unrelated European participants who did not undergo cardiac MRI

		All European Participants
N		313885
Sex		
	Women	173092 (55 %)
	Men	140793 (45 %)
Age at enrollment		
	Mean (SD)	57 (8.0)
	Median [Q1-Q3]	59 [51 - 64]

Ancestry		
	European	313885 (100 %)
BMI (kg/m^2)		
	Mean (SD)	27 (4.8)
	Median [Q1-Q3]	27 [24 - 30]
	Missing	1062 (0.3%)
Height (cm)		
	Mean (SD)	170 (9.3)
	Median [Q1-Q3]	170 [160 - 180]
	Missing	717 (0.2%)
Weight (kg)		
	Mean (SD)	78 (16)
	Median [Q1-Q3]	77 [67 - 88]
	Missing	945 (0.3%)
SBP (mmHg)		
	Mean (SD)	140 (19)
	Median [Q1-Q3]	140 [130 - 150]
	Missing	310 (0.1%)
DBP (mmHg)		
	Mean (SD)	82 (10)
	Median [Q1-Q3]	82 [76 - 89]
	Missing	308 (0.1%)
MET minutes/week		
	Mean (SD)	2700 (3800)
	Median [Q1-Q3]	1500 [580 - 3300]
Standard drinks/week		
	Mean (SD)	12 (11)
	Median [Q1-Q3]	9.3 [5.5 - 16]
	Missing	93280 (29.7%)
Pack year smoking history		
	Mean (SD)	7.1 (15)
	Median [Q1-Q3]	0.0 [0.0 - 7.5]
Smoking status at enrollment		
	Current	33006 (11 %)
	Never	170406 (54 %)
	Prefer not to answer	1108 (0 %)
	Previous	109365 (35 %)
Diabetes diagnosis at baseline		
	Absent	307966 (98 %)
	Present	5919 (2 %)
Hypercholesterolemia diagnosis at baseline		
	Absent	273836 (87 %)
	Present	40049 (13 %)

Hypertension diagnosis at baseline		
	Absent	226196 (72 %)
	Present	87689 (28 %)

This table presents the characteristics of the unrelated participants in the European genetic subset who did not undergo cardiac MRI, and who were assessed for the ability of the polygenic scores to predict incident DCM.

Supplemental Table 24: Polygenic scores from cardiac MRI phenotypes are associated with incident DCM in the European-specific analysis

Source of SNP Score	Hazard Ratio	95% CI Lower	95% CI Upper	P-value
lvedv	1.32	1.18	1.48	2.23E-06
lvedvi	1.23	1.10	1.38	1.91E-04
lvef	0.65	0.59	0.73	1.52E-13
lvesv	1.48	1.32	1.65	7.77E-12
lvesvi	1.56	1.40	1.75	9.10E-15
svi	1.17	1.05	1.30	4.82E-03
svi	1.01	0.91	1.13	8.44E-01

“Hazard Ratio” represents the hazard ratio of a one standard deviation increase in the SNP score on the probability of developing dilated cardiomyopathy.

Supplemental Figure 8: SNP effect estimates in the main analysis and in the European-specific analysis

Each point represents one SNP; the X-coordinate represents the SNP's effect estimate in the main analysis (BETA) and the Y-coordinate represents its effect estimate in the European-

specific analysis (BETA_euro). Perfect correlation would be reflected in a line of $y = x$. All SNPs with $P < 5 \times 10^{-5}$ for each trait are plotted.

Reviewers' Comments:

Reviewer #1:

Remarks to the Author:

I appreciate the authors thorough responses to my comments, in particular to the ancestry issue. I have no further comments and congratulate the authors on a good study and well written manuscript.

NCOMMS-19-24215C

“Analysis of cardiac magnetic resonance imaging traits in 36,000 individuals reveals shared genetic basis with dilated cardiomyopathy”

Reviewer’s Comments:

Reviewer #1 (Remarks to the Author):

I appreciate the authors thorough responses to my comments, in particular to the ancestry issue. I have no further comments and congratulate the authors on a good study and well written manuscript.

Author response:

We appreciate the careful evaluation by the Reviewer and believe that our revised manuscript has been significantly improved as a result of their feedback.